# FOXO1 enhances CAR T cell stemness, metabolic fitness and efficacy

Jack D. Chan[1,2,9], Christina M. Scheffler[1,2,9], Isabelle Munoz[1,2,9], Kevin Sek[1,2], Joel N. Lee[1,2], Yu-Kuan Huang[1,2], Kah Min Yap[1,2], Nicole Y. L. Saw[1,2], Jasmine Li[1,2], Amanda X. Y. Chen[1,2], Cheok Weng Chan[1,2], Emily B. Derrick[1,2], Kirsten L. Todd[1,2], Junming Tong[1,2], Phoebe A. Dunbar[1,2], Jiawen Li[1,2], Thang X. Hoang[1,2], Maria N. de Menezes[1,2], Emma V. Petley[1,2], Joelle S. Kim[1,2], Dat Nguyen[1,2], Patrick S. K. Leung[3], Joan So[2], Christian Deguit[1,2], Joe Zhu[1,2], Imran G. House[1,2], Lev M. Kats[2], Andrew M. Scott[4,5], Benjamin J. Solomon[2], Simon J. Harrison[2,6], Jane Oliaro[1,2], Ian A. Parish[1,2], Kylie M. Quinn[3,7], Paul J. Neeson[1,2], Clare Y. Slaney[1,2], Junyun Lai[1,2,10 ✉], Paul A. Beavis[1,2,10 ✉] & Phillip K. Darcy[1,2,6,8,10 ✉]

Chimeric antigen receptor (CAR) T cell therapy has transformed the treatment of haematological malignancies such as acute lymphoblastic leukaemia, B cell lymphoma and multiple myeloma[1–4], but the efficacy of CAR T cell therapy in solid tumours has been limited[5]. This is owing to a number of factors, including the immunosuppressive tumour microenvironment that gives rise to poorly persisting and metabolically dysfunctional T cells. Analysis of anti-CD19 CAR T cells used clinically has shown that positive treatment outcomes are associated with a more 'stem-like' phenotype and increased mitochondrial mass[6–8]. We therefore sought to identify transcription factors that could enhance CAR T cell fitness and efficacy against solid tumours. Here we show that overexpression of FOXO1 promotes a stem-like phenotype in CAR T cells derived from either healthy human donors or patients, which correlates with improved mitochondrial fitness, persistence and therapeutic efficacy in vivo. This work thus reveals an engineering approach to genetically enforce a favourable metabolic phenotype that has high translational potential to improve the efficacy of CAR T cells against solid tumours.

In the solid tumour microenvironment, CAR T cells are predisposed to terminally differentiate in response to chronic antigen stimulation, metabolic competition and a lack of appropriate co-stimulatory signals[6]. Terminally differentiated CAR T cells are similar to exhausted endogenous T cells that do not eliminate tumours owing to dysfunction, attenuated effector function and poor persistence[9]. Such cells are characterized by reduced expression of effector molecules, the expression of immune checkpoints including PD-1, LAG3 and TIM3 and transcriptional regulators associated with exhaustion, such as IRF4[9–11]. Less-differentiated CAR T cells, defined by a phenotype of CD45RA⁺CD62L⁺, persist for longer[6,8,12]. This is because less-differentiated T cells maintain higher multipotency and greater self-renewal capacity and thus can generate increased numbers of effector-like progeny to facilitate improved tumour control[13,14]. Accordingly, CAR T cells with higher initial frequencies of less-differentiated cells elicit improved persistence and therapeutic potential[8,15].

The processes of T cell differentiation, metabolism and epigenetic reprogramming are highly interdependent[16]. T cell exhaustion is characterized by sub-optimal effector functions that are enforced by epigenetic regulation such as repressive DNA methylation at key gene loci[17], processes that are directly controlled by intracellular metabolites[16]. In the context of CAR T cells, improved oxidative metabolism and increased mitochondrial biogenesis is associated with enhanced persistence and function[7,18]. A variety of approaches have been explored to favourably modulate CAR T cell differentiation. These include the use of homeostatic cytokines, epigenetic regulation, and more recently, the overexpression of transcriptional regulators[6,19–22]. However, none of these genetic reprogramming approaches have identified a transcriptional regulator candidate that can rewire CAR T cells to enhance their metabolic fitness and protect them from exhaustion. We therefore sought to identify key transcription factors that are upregulated by IL-15, given that IL-15 enhances CAR T cell persistence and metabolism[12,23]. Analysis of the epigenome and transcriptome of CAR T cells cultured in the presence of IL-15 revealed a strong enrichment of a *Foxo1* gene signature. Overexpression of a constitutively active variant of FOXO1 (FOXO1-ADA) in mouse HER2-specific CAR T cells or wild-type FOXO1 in human Lewis Y CAR T cells improved metabolic fitness, ultimately leading to enhanced control of solid tumours. Notably, FOXO1 overexpression maintained CAR T cell 'stemness' and enhanced persistence to a greater extent than other transcriptional regulators

[1]Cancer Immunology Program, Peter MacCallum Cancer Centre, Melbourne, Victoria, Australia. [2]Sir Peter MacCallum Department of Oncology, The University of Melbourne, Parkville, Victoria, Australia. [3]School of Health and Biomedical Sciences, RMIT University, Bundoora, Victoria, Australia. [4]Olivia Newton-John Cancer Research Institute and School of Cancer Medicine, La Trobe University, Melbourne, Victoria, Australia. [5]Faculty of Medicine, The University of Melbourne, Parkville, Victoria, Australia. [6]Clinical Haematology and Centre of Excellence for Cellular Immunotherapies, Peter MacCallum Cancer Centre and Royal Melbourne Hospital, Melbourne, Victoria, Australia. [7]Department of Biochemistry and Molecular Biology, Monash University, Clayton, Victoria, Australia. [8]Department of Immunology, Monash University, Clayton, Victoria, Australia. [9]These authors contributed equally: Jack D. Chan, Christina M. Scheffler, Isabelle Munoz. [10]These authors jointly supervised this work: Junyun Lai, Paul A. Beavis, Phillip K. Darcy. ✉e-mail: junyun.lai@unimelb.edu.au; paul.beavis@petermac.org; phil.darcy@petermac.org

upregulated by IL-15, such as TCF7 (also known as TCF1) and ID3. FOXO1 overexpression implemented a broad transcriptional and epigenetic programme that led to an enrichment of signatures associated with persistence, but did not preclude FOXO1-expressing CAR T cells from acquiring robust effector function upon antigen stimulation. Overall, our findings show that overexpression of FOXO1 enhances CAR T cell efficacy in the treatment of solid tumours and provides promising groundwork to utilize this approach in a clinical setting.

## IL-15 transcriptome identifies FOXO1

Previous studies by both our group and others have shown that CAR T cells generated with IL-7 and IL-15 elicit greater long-term persistence relative to CAR T cells cultured in IL-2 and IL-7. This effect is linked to their improved metabolic fitness[12,23]. Indeed, CAR T cells cultured with IL-15 exhibited a more pronounced central memory T ($T_{CM}$) cell-like phenotype (CD62L$^+$CD44$^+$) and improved persistence in mice bearing E0771 HER2$^+$ tumours (Extended Data Fig. 1a,b). To interrogate the transcriptional regulation of this phenotype, we compared the epigenome and transcriptome of CD8$^+$ CAR T cells cultured in the presence of IL-15 and IL-2. RNA-sequencing (RNA-seq) analysis revealed increased expression of genes associated with memory in IL-15 generated CAR T cells including *Jun*, *Tcf7*, *Id3*, *Foxo1* and *Klf2*, a known FOXO1 target gene[24,25] (Extended Data Fig. 1c,d). Furthermore, IL-15 induced a downregulation of glycolytic genes, consistent with the notion that IL-15 induces a favourable metabolic phenotype[23] (Extended Data Fig. 1e). Notably, analysis of IL-15-induced transcriptional changes revealed that FOXO1 target genes were the most significantly enriched subset, based on both a publicly available chromatin immunoprecipitation with sequencing (ChIP–seq) dataset and a list of genes with at least one canonical FOXO1-binding motif within 2 kb either side of the transcriptional start site (Extended Data Fig. 1f,g). At the epigenetic level, FOXO1 was also one of the highest-ranking transcription factors. Assay for transposase-accessible chromatin with sequencing (ATAC-seq) analysis revealed a significant enrichment for FOXO1-binding motifs in genomic regions that became more accessible after culture in IL-15 as determined by both Homer (Extended Data Fig. 1h, i) and ChromeVAR analyses (Extended Data Fig. 1j). Of note, motif enrichment was also observed for FLI1, which has previously been shown to negatively regulate T cell persistence through antagonism of ETS–RUNX activity, TCF7, a factor known to regulate memory formation, and TCF3, a transcription factor that is known to interact with ID3[11,26,27]. Given these data, we interrogated the role of FOXO1 in CAR T cell function using CRISPR–Cas9-mediated targeting. Consistent with a role for FOXO1 in maintaining CAR T cells in a favourable differentiation state, deletion of FOXO1 in mouse CAR T cells led to a significant reduction in CD62L and TCF7 expression, and upon serial co-culture with E0771-HER2 tumour cells led to significantly increased expression of TIM3, PD-1 and LAG3, altogether indicative of a transition to a more short-term effector-like phenotype (Extended Data Fig. 1k–m).

## FOXO1 enhances CAR T cell functionality

Given the strong molecular signature of FOXO1 in IL-15-conditioned CAR T cells and the critical importance of FOXO1 in maintaining CAR T cell stemness, we next investigated the phenotype and function of FOXO1-overexpressing CAR T cells. Because T cell activation facilitates the exclusion of wild-type FOXO1 from the nucleus via post-translational modifications, we elected to overexpress FOXO1-ADA, a constitutively active variant of FOXO1, in mouse anti-HER2 CAR T cells[28,29]. We compared FOXO1-ADA-overexpressing CAR T cells with CAR T cells overexpressing TCF7, ID3 and JUN, given previous data indicating the enhanced efficacy of JUN-overexpressing CAR T cells[22]. Successful transduction of T cells was confirmed by expression of MYC-tagged

CAR and overexpression of the respective factors using quantitative PCR with reverse transcription of sorted CD8$^+$NGFR$^+$ CAR T cells (Extended Data Fig. 2a,b). The effects of transcription factor overexpression were assessed in vitro via a re-stimulation assay, which involved re-stimulating CAR T cells with E0771-HER2 tumours three times before phenotypic and functional analyses (Extended Data Fig. 3a).

Conventional CAR T cells lose their capacity to secrete IFNγ and TNF over multiple rounds of antigenic stimulation owing to exhaustion or dysfunction. This limits their efficacy, given that these cytokines are required for their anti-tumour function[30–33]. The overexpression of TCF7 and FOXO1-ADA led to significantly increased IFNγ and TNF production relative to control CAR T cells after each successive round of stimulation (Extended Data Fig. 3b,c). Moreover, FOXO1-ADA, and to a lesser extent JUN or TCF7 overexpression led to significantly increased expression of GZMB, suggestive of prolonged killing capacity (Extended Data Fig. 3d,e). Notably, the overexpression of FOXO1-ADA, TCF7, ID3 or JUN led to a significant increase in CAR T cell numbers and maintenance of a less-differentiated CAR T cell surface phenotype, as indicated by the total number of CD8$^+$ CAR T cells including those exhibiting a CD62L$^+$Ly108$^+$ stem-like phenotype (Extended Data Fig. 3f).

## FOXO1 boosts CAR T cell in vivo efficacy

Having established that overexpression of pro-memory transcription factors enhanced CAR T cell polyfunctionality in vitro, we next investigated their in vivo anti-tumour potential. We used an immunocompetent, transgenic C57BL/6 HER2 mouse model that expresses human HER2 in breast epithelial and brain tissue. In mice bearing orthotopic E0771-HER2 breast tumours or subcutaneous MC38-HER2 tumours, the overexpression of TCF7, FOXO1-ADA or ID3, but not JUN, significantly enhanced CAR T cell efficacy through more durable anti-tumour control, and a significant increase in survival for mice treated with FOXO1-ADA or ID3-expressing CAR T cells relative to controls (Fig. 1a–c and Extended Data Fig. 4a–d). Notably, the overexpression of wild-type FOXO1 did not significantly enhance the therapeutic efficacy, indicating that the constitutively active form was necessary to improve the function of mouse CAR T cells (Extended Data Fig. 4e).

We next investigated the in vivo phenotype of CAR T cells isolated from tumours and draining lymph nodes (dLNs) of tumour-bearing mice nine days after treatment. CAR T cells overexpressing FOXO1-ADA exhibited significantly increased production of the pro-inflammatory cytokines IFNγ and TNF relative to control CAR T cells (Fig. 2a). Indeed, FOXO1-ADA most significantly improved polyfunctionality as indicated by a higher frequency of CAR T cells expressing two or more effector molecules (IFNγ, TNF or GZMB), and was the only factor that resulted in significantly fewer cells expressing none of the analysed effector molecules relative to control CAR T cells (Fig. 2b). Assessment of other parameters of CD8$^+$ CAR T cell function revealed that FOXO1-ADA overexpression led to only a modest increase in Ki-67 expression both within the total CD8$^+$ CAR T cell population, and the TCF7$^+$ less-differentiated subset (Extended Data Fig. 5a,b). Similarly, FOXO1-ADA-expressing CAR T cells also expressed similar levels of the activation markers PD-1 and TIM3 relative to control CAR T cells, consistent with the notion that FOXO1-ADA expression did not prevent CAR T cells from acquiring an effector-like phenotype upon tumour antigen recognition (Extended Data Fig. 5c). Given the expression of PD-1 on these CAR T cells, we investigated whether therapeutic benefit could be augmented through the addition of immune checkpoint blockade therapy. Indeed, combination of FOXO1-ADA expressing CAR T cells with anti-PD-1 resulted in significantly improved tumour growth inhibition compared with control CAR T cells, suggesting synergy between FOXO1-ADA CAR T cells and PD-1 blockade (Extended Data Fig. 5d). Next, we assessed the effects of transcription factor expression on the number of CD4$^+$ CAR T cells. Consistent with our previous observations[12], CD4$^+$ CAR T cells made up only a minor proportion of tumour-infiltrating CAR T cells and this was

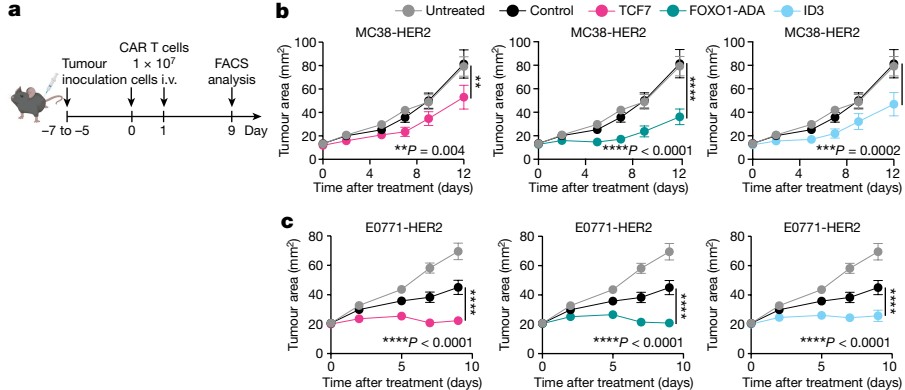

**Fig. 1 | FOXO1-ADA overexpression enhances therapeutic activity of CAR T cells. a**, Schematic for in vivo experiments. i.v., intravenous injection. **b,c**, Treatment of subcutaneous MC38-HER2 (**b**) or mammary fat pad E0771-HER2 (**c**) tumours. Tumours were established in mice for 5 to 7 days prior to treatment with two doses of 1 × 10⁷ indicated CAR T cells administered on subsequent days. Tumour growth is represented as mean tumour size of $n = 10$ (control), $n = 11$ (TCF7, FOXO1-ADA) and $n = 12$ (untreated) (**b**) or $n = 17$ (untreated) and $n = 18$ (control, TCF7, FOXO1-ADA, ID3) (**c**) mice per group ± s.e.m. from 3 pooled experiments. Two-way ANOVA. $*P < 0.05$, $**P < 0.01$, $***P < 0.001$, $****P < 0.0001$.

not significantly modulated by expression of TCF7, ID3 or FOXO1-ADA (Extended Data Fig. 5e).

Secretion of IFNγ has been shown to be critical for CAR T cell anti-tumour efficacy in the solid tumour setting[30,33–35]. Given that FOXO1-ADA expression led to enhanced cytokine production, we assessed the contribution of IFNγ to the therapeutic activity of FOXO1-ADA-overexpressing CAR T cells. These experiments revealed that IFNγ blockade abrogated the enhanced therapeutic activity of FOXO1-ADA-overexpressing CAR T cells, indicating that IFNγ produced by FOXO1-ADA-expressing cells is likely to have led to the observed enhanced therapeutic effects (Fig. 2c).

To interrogate potential effects on metabolic function, we first investigated the effect of transcription factor expression on mitochondrial mass and fitness within intratumoral CD8⁺ CAR T cells. CAR T cells isolated from the tumours of treated mice were stained ex vivo with the mitochondrial dyes Mitotracker Deep Red (MDR) and Mitotracker Green (MG), which are indicative of properly regulated mitochondrial membrane potential and mitochondrial mass, respectively. Compared with control CAR T cells, intratumoral CAR T cells overexpressing TCF7 or FOXO1-ADA, but not ID3, displayed an increased ratio of cells with functional mitochondria (designated as MDR^hiMG^mid/hi) relative to dysfunctional mitochondria (MDR^midMG^hi). This was also reflected by higher MDR and MG staining intensities, indicative of properly regulated membrane potential and mitochondrial mass[36] (Fig. 2d,e). Furthermore, RNA-seq of in vitro-stimulated CAR T cells revealed that an oxidative phosphorylation signature was among the most significantly upregulated pathways in FOXO1-ADA expressing CAR T cells relative to control CAR T cells (Fig. 2f and Extended Data Fig. 6). Notably, increased oxidative phosphorylation has been associated with the increased accumulation of MDR^hiMG^mid/hi cells[37], and thus this observation is consistent with in vivo phenotypes observed with FOXO1-ADA expressing CAR T cells. Collectively, our data suggest that overexpression of FOXO1-ADA in CAR T cells contributes to enhanced in vivo anti-tumour activity by promoting T cell polyfunctionality, proliferation and mitochondrial biogenesis to support effector functions.

As FOXO1, ID3 and TCF7 contribute to the formation of memory T cells that home to secondary lymphoid tissues, we investigated whether CAR T cells overexpressing transcriptional regulators would exhibit enhanced trafficking to tumour dLNs. Indeed, quantification of CD8⁺ CAR T cells within dLNs revealed that the overexpression of each of these factors increased the number of CAR T cells residing in tumour dLN relative to control CAR T cells (Fig. 2g). This was also associated with an increased presence of T memory stem cell (T_SCM)-like cells (CXCR3⁺CD62L⁺) relative to control CAR T cells (Fig. 2h). Notably,

FOXO1-ADA significantly enhanced the number of CAR T cells at the dLNs but not at the non-draining lymph nodes from the opposite flank (Fig. 2i). This suggests that FOXO1-ADA expression specifically enhanced the migration and/or expansion of CAR T cells at the dLN. To investigate the importance of dLN residency to the phenotype of FOXO1-ADA expressing CAR T cells, we correlated the number of dLN-resident CD8⁺ T cells to the frequency of intratumoral IFNγ⁺TNF⁺ cells in each treatment group and observed a significant positive correlation in mice treated with TCF7 and FOXO1-ADA-overexpressing CAR T cells (Fig. 2j). Collectively, our data infer that CAR T cell trafficking through tumour dLNs may be promoted through the overexpression of FOXO1-ADA and contributes to improved CAR T cell polyfunctionality. The increased therapeutic efficacy observed with FOXO1-ADA-overexpressing CAR T cells was not associated with overt signs of toxicity (Extended Data Fig. 4f–i), highlighting the clinical potential of this approach.

## FOXO1 improves human CAR T cell stemness

We next investigated the potential of FOXO1 overexpression in the context of human CAR T cells. We transduced human T cells with Lewis Y CAR (which is currently being used in a phase 1b clinical trial (ClinicalTrials.gov ID NCT03851146)) linked to FOXO1-ADA via a P2A peptide (Extended Data Fig. 7a). The FOXO1-ADA-overexpressing CAR T cells appeared less differentiated, as indicated by an increased frequency of CD62L⁺CD27⁺ cells and CD45RA⁺ cells concomitant with reduced expression of the exhaustion markers LAG3 and TIM3 (Extended Data Fig. 7b,c), a phenotype previously associated with improved CAR T cell responses in the clinic[8,15]. However, unlike the case with mouse CAR T cells, FOXO1-ADA overexpression prevented human CAR T cells from acquiring an effector-like phenotype after activation. Although FOXO1-ADA-overexpressing CAR T cells significantly upregulated CD69 upon co-culture with tumour cells, (Extended Data Fig. 7d), their capacity to produce IFNγ and TNF was significantly attenuated relative to control CAR T cells (Extended Data Fig. 7e). This suggested that expression of constitutively active FOXO1 restricted the capacity of CAR T cells to gain full effector function and therefore led us to evaluate the overexpression of wild-type FOXO1 in human CAR T cells, which we hypothesized could result in a more stem-like phenotype while enabling effective transition to effector-like cells in response to antigen stimulation. Indeed, whereas overexpression of wild-type FOXO1 significantly enhanced the proportion of CD45RA⁺ and CD62L⁺CD27⁺ CAR T cells (Fig. 3a)−in contrast to FOXO1-ADA expressing CAR T cells−wild-type FOXO1-expressing CAR T cells were able to produce similar levels of

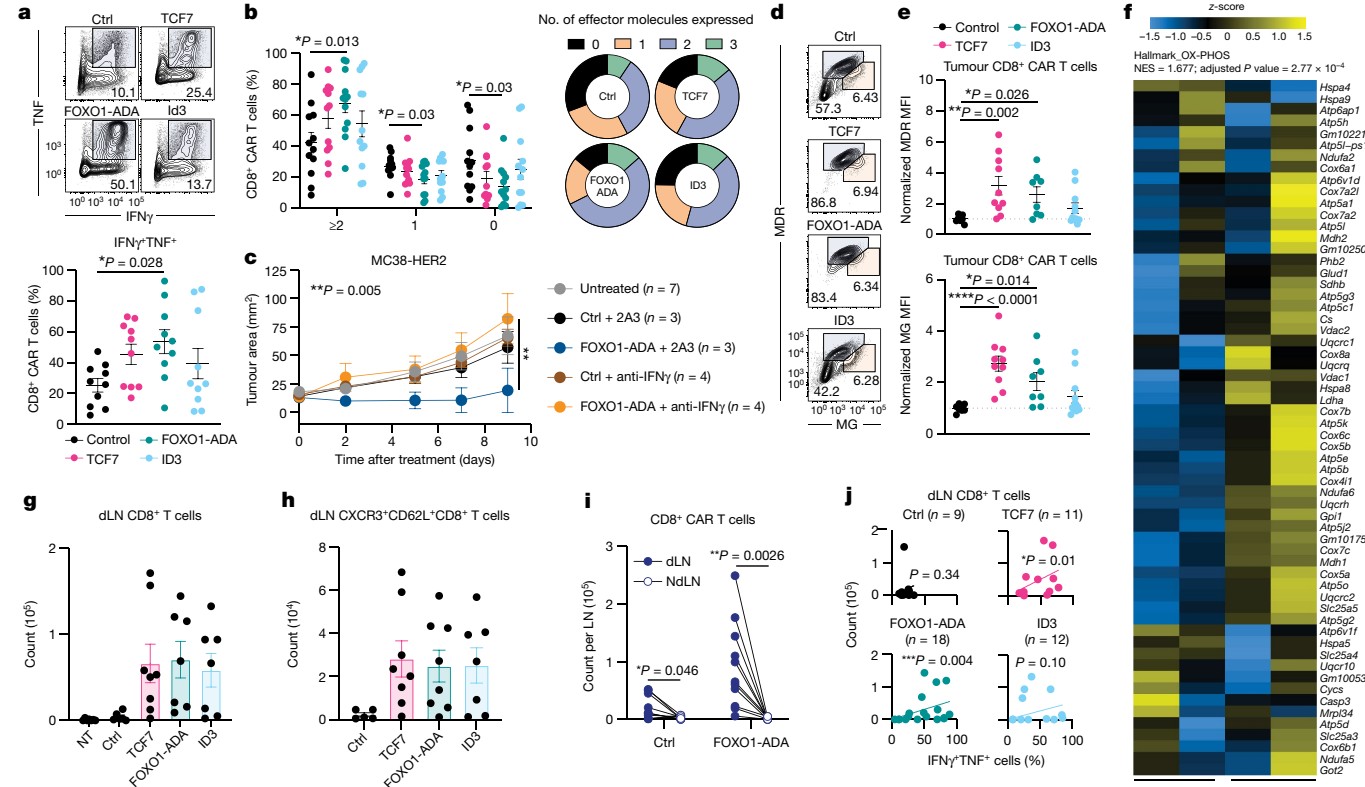

**Fig. 2 | FOXO1-ADA overexpression enhances in vivo polyfunctionality and metabolic fitness. a,b,d,e,g–j,** Tissues from E0771-HER2 tumour-bearing mice were analysed nine days after treatment. **a,b,d,e,** Flow cytometry analysis of tumour-infiltrating CAR T cells. 2A3, isotype control antibody; ctrl, control; MFI, mean fluorescence intensity. **a,** Frequency of tumour-infiltrating CD8[+] CAR T cells expressing IFNγ and TNF. Top, combined samples. Bottom, data from individual mice. **b,** Frequency of CAR T cells expressing 0, 1, 2 or 3 effector proteins (IFNγ, TNF or GZMB). **c,** MC38-HER2 tumour growth in mice treated as in Fig. 1. Indicated mice were co-treated with anti-IFNγ. **d,e,** MDR and MG staining of CD8[+] CAR T cells. **d,** Combined samples. **e,** Individual mice. **f,** Heat map for genes in the oxidative phosphorylation Hallmark pathway of CD8[+] CAR T cells at 72 h after anti-CAR stimulation. NES, normalized enrichment score.

**g–j,** Flow cytometry analysis of CAR T cells associated with tumour, dLN or non-draining lymph node (ndLN). Number of total (**g**) or CXCR3[+]CD62L[+] (**h**) tumour dLN-resident CD8[+] CAR T cells. **i,** Paired analysis of CAR T cell numbers in the ndLN and dLN of *n* = 11 mice per group. LN, lymph node. **j,** Correlation of number of tumour dLN-resident CD8[+] CAR T cells with frequency of IFNγ[+]TNF[+] intratumoral CD8[+] CAR T cells. **a,b,e,g,h,** Data are mean ± s.e.m. from *n* = 10 (**a**), *n* = 12 (**b**), *n* = 8 (control, FOXO1-ADA), *n* = 10 (TCF7) or *n* = 11 (ID3) (**e**), *n* = 7 (non-treatment (NT), FOXO1-ADA, ID3), *n* = 6 (control), *n* = 8 (TCF7) (**g**), *n* = 5 (control), *n* = 8 (TCF7, FOXO1-ADA), *n* = 7 (ID3) (**h**) mice pooled from two independent experiments. Two-sided, one-way ANOVA. **c,** Data are mean ± s.e.m. of indicated number of mice. Two-way ANOVA. **i,** Two-sided, paired *t*-test. **j,** Number of mice as indicated. Simple linear regression analysis.

IFNγ and TNF to control CAR T cells upon co-culture with tumour cells (Extended Data Fig. 7e). We hypothesized that the phenotypic differences observed between mouse and human CAR T cells following overexpression of the constitutively active variant of FOXO1 may be related to the more efficient upregulation of FOXO1 observed in human cells (Extended Data Fig. 2b,c). To test this, we modified the human lentiviral vector such that expression of FOXO1 was driven by a *PGK* or CMV promoter, both of which have been reported to give lower transgene expression relative to the EF1α (also known as *EEF1A1*) promoter[38]. These experiments revealed that the CMV promoter induced a lower level of FOXO1 expression (Extended Data Fig. 7f), which resulted in a reduced increase in the CD45RA[+]CD62L[+] population relative to control CAR T cells (Extended Data Fig. 7g). However, FOXO1-ADA significantly attenuated the production of IFNγ and TNF by CAR T cells regardless of the promoter used to drive its expression (Extended Data Fig. 7h), and given these results, we proceeded with evaluation of the effect of wild-type FOXO1 expression driven by the EF1α promoter.

We next compared the effects of FOXO1, TCF7 and ID3 on the phenotype of Lewis Y CAR T cells. FOXO1 was significantly more effective in the induction of the CD45RA[+]CD62L[+] population of CD8[+] CAR T cells relative to TCF7 and ID3 (Fig. 3b), and indeed the increased proportion of CD45RA[+]CD62L[+] cells following FOXO1 expression was reproducible across multiple donors (Fig. 3c). FOXO1 overexpression was

also associated with increased expression of CCR7 and CX3CR1, and reduced expression of CD39, TIM3 and PD-1 (Fig. 3d and Extended Data Fig. 8a). In the context of serial co-cultures with OVCAR-3 or MCF7 tumour cells, FOXO1 overexpression led to a significant increase in the recovery of CD8[+] and CD4[+] CAR T cells and similar levels of IFNγ and TNF production compared with control CAR T cells (Extended Data Fig. 8b,c). Phenotypic analysis of FOXO1-overexpressing CAR T cells following co-culture with tumour cells revealed a similar expression of PD-1 and TIM3, but reduced expression of CD39 relative to control CAR T cells (Extended Data Fig. 8d,e). Notably, the reduced expression of CD39 mediated by FOXO1 was not recapitulated by the overexpression of either TCF7 or ID3 in T cell-only cultures (Extended Data Fig. 8f) or following co-culture with tumour cells (Extended Data Fig. 8g), providing further evidence for an enhanced capacity of FOXO1 to favourably modulate CAR T cell phenotype relative to these factors. In reverse experiments, CRISPR–Cas9-mediated targeting of FOXO1 in human Lewis Y (LeY) CAR T cells led to a loss of CD62L expression and a reduction in CAR T cell expansion upon extended culture (Extended Data Fig. 9a–d), and impaired production of IFNγ upon co-culture with tumour cells (Extended Data Fig. 9e). Given that phenotypic changes in FOXO1-expressing CAR T cells were largely reversible upon CAR T cell activation, we interrogated whether FOXO1-expressing CAR T cells could recover a more stem-like phenotype after transient activation

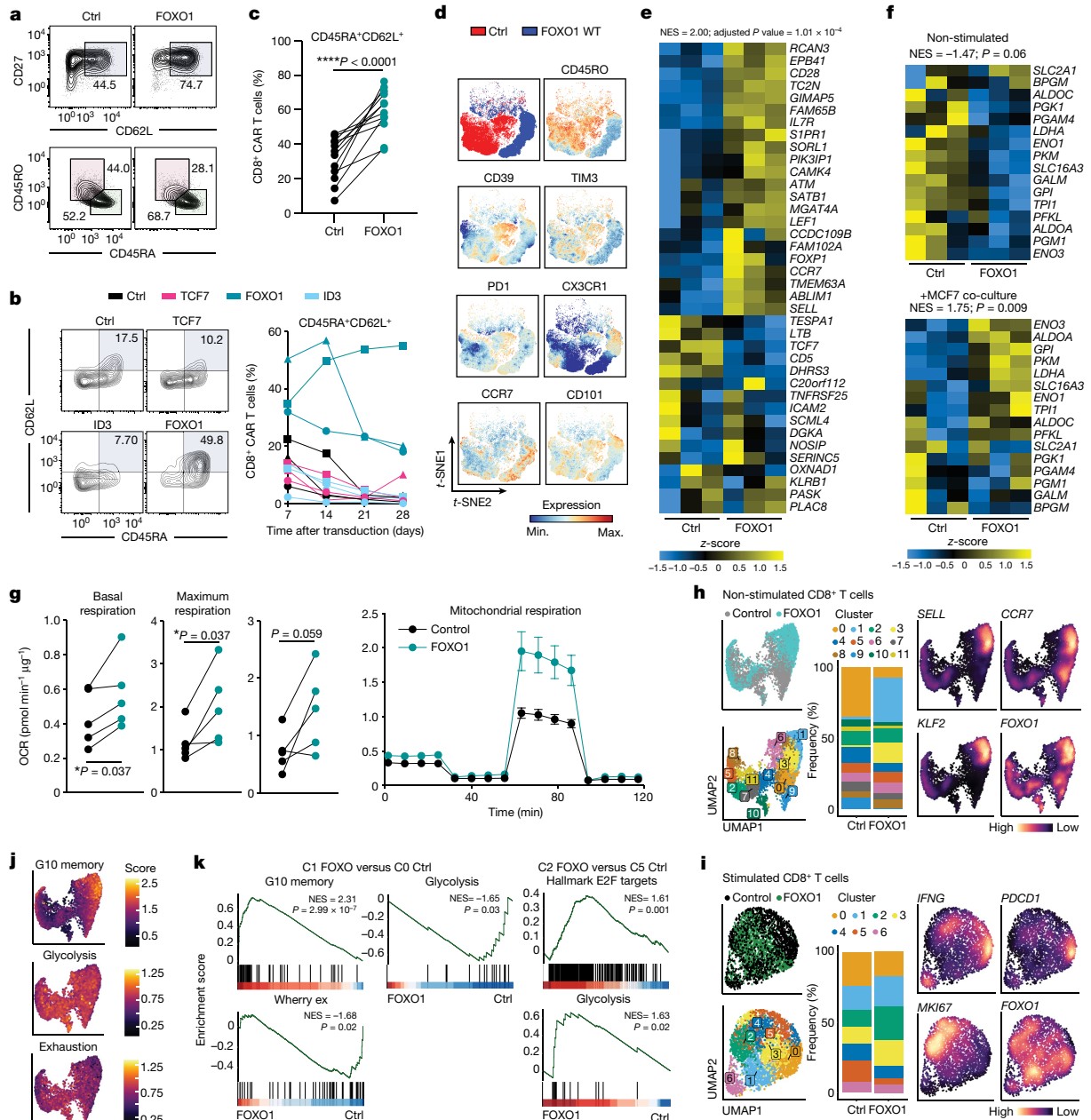

**Fig. 3 | FOXO1 overexpression enhances the stemness and metabolic fitness of human CAR T cells.** Analysis of human CD8[+]CAR[+] T cells expressing mCherry (ctrl), FOXO1, TCF7 or ID3. **a**, Proportion of CD62L[+]CD27[+] or CD45RA[+] CAR T cells. **b**, Proportion of CD8[+]CAR[+] T cells with a CD45RA[+]CD62L[+] phenotype. Left, representative staining (day 14 post-transduction). Right, time course analysis. Different shapes represent individual donors. **c**, Paired analysis from 12 individual experiments. **d**, *t*-Distributed stochastic neighbour embedding (*t*-SNE) plot of indicated cell surface markers. Max., maximum; min., minimum; WT, wild type. **e**,**f**, CD8[+]CAR[+] T cells were analysed by RNA-seq before and after activation with MCF7 tumour cells. *n* = 3 biological replicates. **e**, Heat map depicting the 38 genes with highest differential expression in the G10 memory cluster identified by Sade-Feldmann et al.[11]. **f**, expression of glycolysis-related genes before and after co-culture with MCF7 tumour cells. **g**, Analysis of CAR T cell oxidative consumption by Seahorse MitoStress assay. Data represent

paired analysis from *n* = 5 independent donors (left) or mean ± s.e.m. of *n* = 3 technical replicates of a representative donor (right). OCR, oxygen consumption rate. **h**–**k**, Control or FOXO1-expressing CAR T cells were left unstimulated or were stimulated for 16 h with MCF7 tumour cells and then analysed by single-cell RNA-seq (scRNA-seq). **h**,**i**, Uniform manifold approximation and projection (UMAP) plots, cell cluster composition and density plots showing expression of indicated genes of non-stimulated (**h**) and stimulated (**i**) CD8[+]CAR T cells. **j**, Visualization of gene signatures scores (SingleCellSignature) of memory, glycolysis and exhaustion gene sets in non-stimulated CD8[+] CAR T cells. **k**, Gene set enrichment analysis for indicated pathways comparing FOXO1-expressing CAR T cells in cluster 1 to control CAR T cells within cluster 0 (non-stimulated clusters) or FOXO1-expressing CAR T cells in cluster 2 to control CAR T cells in cluster 5 (stimulated clusters). Two-sided, paired *t*-test (**c**,**g**).

and recovery or in long-term chronic stimulation assays. Indeed, FOXO1-expressing CAR T cells maintained a less-differentiated state after activation followed by one week of rest (Extended Data Fig. 10a), or when CAR T cells were exposed to tumour cells over a period of three weeks (Extended Data Fig. 10b).

## Profiling FOXO1 CAR T cell transcriptome

To further understand the mechanisms underlying the more stem-like phenotype of FOXO1-overexpressing CAR T cells, we analysed CD8[+] and CD4[+] CAR T cells using RNA-seq. This analysis revealed that

FOXO1-overexpressing, but not TCF7-overexpressing, CD8[+] CAR T cells exhibited increased expression of genes associated with less-differentiated T cells relative to their control counterparts (Fig. 3e and Extended Data Fig. 8h). Moreover, FOXO1-overexpressing CD8[+] CAR T cells displayed decreased expression of immune checkpoints relative to control counterparts and this effect was also more marked than with TCF7-expressing CAR T cells (Extended Data Fig. 8i,j). Given our previous observations of improved mitochondrial fitness in mouse FOXO1-ADA-overexpressing CAR T cells, we assessed the effect of wild-type FOXO1 overexpression on genes associated with metabolic pathways both before and after overnight co-culture with MCF7 tumour cells. In the steady state, both CD8[+] and to a lesser extent CD4[+] FOXO1-overexpressing CAR T cells exhibited reduced expression of glycolytic genes, consistent with a more stem-like phenotype and as observed with IL-15 cultured CAR T cells. However, we observed a switch in this phenotype after activation, whereby FOXO1-overexpressing CD8[+] CAR T cells showed higher expression of glycolytic genes relative to control CAR T cells following co-culture with Lewis Y[+] tumour cells (Fig. 3f and Extended Data Fig. 8k). Indeed, we observed that after activation the number of differentially expressed genes (DEGs) between control and FOXO1-expressing CAR T cells was significantly reduced, highlighting that expression of FOXO1 enabled the acquisition of a more stem-like phenotype without preventing the acquisition of effector function (Extended Data Fig. 8l). To further investigate the metabolic phenotype of FOXO1-overexpressing CAR T cells, we performed a Seahorse analysis. FOXO1-expressing CAR T cells exhibited a significantly enhanced basal respiratory rate and maximal respiratory rate, with an increase in spare respiratory capacity also observed in 4 out of 5 tested donors (Fig. 3g).

Furthermore, FOXO1 overexpression resulted in a significant negative enrichment of genes associated with exhaustion[39] relative to control CAR T cells, an effect that was not observed following TCF7 overexpression (Extended Data Fig. 8m). We next evaluated this in the context of stimulation and resting. In this repeat experiment, we again observed that prior to stimulation FOXO1-expressing CAR T cells exhibited reduced expression of exhaustion-related genes (Extended Data Fig. 8n). Following stimulation through the CAR, both control and FOXO1-expressing CAR T cells upregulated these genes such that there was no significant difference between the groups. However, after a period of rest, the negative enrichment for this gene set was restored in FOXO1-expressing CAR T cells, highlighting that these cells exhibit durable protection from the transcription of genes associated with exhaustion.

To further understand the effect of FOXO1 overexpression on CAR T cell phenotype, we performed scRNA-seq analyses on CAR T cells prior to and after co-culture with MCF7 tumour cells. Unbiased clustering analysis confirmed that, consistent with the bulk RNA-seq data, there were significant differences between control and FOXO1-expressing CAR T cells but that these differences were diminished after activation (Extended Data Fig. 11a,b). We therefore further dissected our analysis into non-activated CD8[+] CAR T cells (Fig. 3h) and activated CD8[+] CAR T cells (Fig. 3i). This confirmed that FOXO1 expression enhanced the proportion of CD8[+] CAR T cells with a more stem-like phenotype, notably through the enrichment of a cluster (designated cluster 1) that was characterized by high expression of KLF2, SELL and IL7R (Fig. 3h and Extended Data Fig. 11c). This enhanced proportion of cluster 1 cells was offset by a reduction in the proportion of a cluster of cells (designated cluster 0) characterized by high expression of IFITM3 and TNFSF10 (Extended Data Fig. 11c). Indeed, comparing FOXO1-expressing CAR T cells in cluster 1 to control CAR T cells in cluster 0 revealed a significant enrichment for genes associated with less-differentiated cells and reduced expression of genes associated with glycolysis and exhaustion, corroborating our results from bulk RNA-seq (Fig. 3j,k). Moreover, cluster 1 cells expressed high levels of CCR7, suggesting that these cells have the potential to traffic to lymph nodes, consistent with the phenotype

observed with mouse FOXO1-ADA expressing CAR T cells (Fig. 2g–i). Similar results were observed with non-activated CD4[+] T cells with enrichment of a cluster that exhibited high expression of SELL, KLF2 and IL7R (designated cluster 3) (Extended Data Fig. 11b,d). To further interrogate the phenotype of CAR T cells post-activation, we repeated the analysis on only the activated cells. In this context, we observed six distinct clusters of CD8[+] CAR T cells. Although differences between control and FOXO1-expressing CAR T cells were less marked than prior to activation, this analysis revealed that FOXO1 overexpression led to an enrichment of a cluster (designated cluster 2) characterized by high expression of MKI67 and a concomitant decrease in cluster 5 (Fig. 3i). Comparison of the gene-expression profiles of FOXO1-expressing cluster 2 cells and control cluster 5 cells revealed a positive enrichment for E2F target genes and glycolysis-related genes, suggesting that FOXO1-expressing CAR T cells are primed for a proliferative burst post-activation (Fig. 3i).

## Epigenetic analysis of FOXO1 CAR T cells

We next investigated the effect of FOXO1 expression on the epigenetic landscape of CD8[+] CAR T cells. FOXO1 overexpression led to marked changes, with 3,653 peaks showing increased accessibility and 14,335 peaks with decreased accessibility (Fig. 4a). In line with our observations of the effect of FOXO1 on the expression of immune checkpoints at both the protein and transcriptional level (Fig. 3d and Extended Data Fig. 8a,i), we observed that FOXO1 expression led to decreased accessibility at regions in proximity to the transcriptional start sites of genes encoding negative immune checkpoints such as PDCD1, ENTPD1 and TIGIT (Fig. 4b). ChromVAR analysis revealed that regions that were significantly more accessible in FOXO1-expressing CAR T cells were enriched for motifs that included several other forkhead box family members, STAT1, STAT3 and CTCF, a factor that has been shown to regulate CD8[+] T cell homeostasis through its interaction with TCF7[40]. Compared with control CAR T cells, FOXO1 overexpression strongly reduced the accessibility in regions containing motifs associated with AP-1 and NF-κB subunit binding, in line with previous observations indicating that FOXO1 promotes a resting T cell state through repression of AP-1 activity[41] (Extended Data Fig. 12a). Indeed, FOXO1 overexpression resulted in reduced expression of transcription factors previously observed to be upregulated in exhausted CAR T cells[22], including members of the AP-1 family such as IRF8 and FOSB (Extended Data Fig. 12b). In line with our hypothesis that FOXO1 expression imparts a phenotype akin to IL-15 cultured cells, there was a clear correlation for motif accessibility between IL-15 cultured and FOXO1-expressing CAR T cells (Fig. 4c). Although FOXO1 implemented epigenetic changes consistent with reprogramming to a more stem-like phenotype, it did not prevent the acquisition of an effector-like epigenetic landscape post-activation (Fig. 4d). Indeed, after stimulation through the CAR, FOXO1-overexpressing CAR T cells exhibited only 2,533 sites with increased accessibility and 5,217 with decreased accessibility relative to control CAR T cells, a decrease of around 30% and 65%, respectively, compared to prestimulated samples (Fig. 4e). The sites that exhibited reduced accessibility in resting FOXO1-expressing CAR T cells but not activated FOXO1-expressing CAR T cells (relative to control CAR T counterparts) were enriched in the promoter regions of genes at less than 1 kb from the transcriptional start site (Extended Data Fig. 12c). Thus, accessibility to key effector genes such as IFNG, TNF and IL2 were similar in control and FOXO1-expressing cells after activation (Fig. 4b) and the decreased accessibility of sites of BATF−AP-1 motifs observed in resting FOXO1-expressing CAR T cells was no longer apparent after activation (Extended Data Fig. 12a). Notably, and consistent with our functional data, this suggests that expression of wild-type FOXO1 promotes a stem-like phenotype, but does not preclude epigenomic reprogramming required for T cell activation. Together, these in vitro observations highlighted the potential for FOXO1 overexpression to

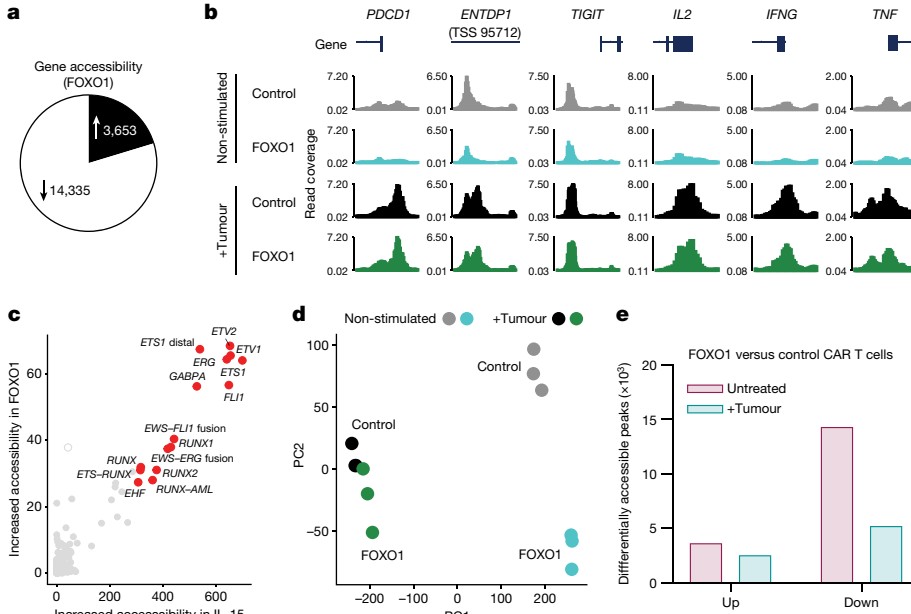

**Fig. 4 | FOXO1 overexpression induces an epigenetic landscape that promotes CAR T cell stemness but does not preclude effector-like transition upon CAR T cell activation.** Human FOXO1 or control CAR T cells generated as in Fig. 3 were analysed by ATAC-seq 7 days after generation with no stimulation or after 16 h co-culture with MCF7 tumour cells at a 1:1 ratio. CD8+ CAR T cells were purified by fluorescence-activated cell sorting (FACS) prior to analysis. The experiment was performed in *n* = 3 biological replicates. **a**, Differential peak analysis of non-stimulated control or FOXO1-expressing CAR T cells. **b**, IGV tracks for indicated genes in indicated CAR T cell groups. **c**, Correlation of motifs upregulated by IL-15 relative to IL-2 in mouse CAR T cells as in Extended Data Fig. 1 and in FOXO1-overexpressing CAR T cells as determined by HOMER analysis. **d**, Principal component analysis of ATAC-seq data for indicated CAR T cell populations. **e**, Number of peaks with differential accessibility in FOXO1-expressing CAR T cells relative to controls before and after stimulation.

enhance the activity of human CAR T cells and led us to investigate this in the context of treatment of solid tumours.

## FOXO1 augments human CAR T cell therapy

To compare the effect of TCF7, ID3 and FOXO1 overexpression in vivo we treated OVCAR-3 tumour-bearing mice with Lewis Y CAR T cells. FOXO1 overexpression enhanced CAR T cell efficacy as indicated by a significant decrease in tumour mass (Fig. 5a), which was associated with a significant increase in the number of CD8+ and CD4+ CAR T cells in the blood and spleen of treated mice (Fig. 5b,c). Indeed, overexpression of FOXO1, but not TCF7, significantly enhanced control of tumour growth relative to control CAR T cells (Fig. 5d and Extended Data Fig. 13a,b). FOXO1 overexpression significantly increased the proportion of CD8+ CAR T cells in the tumours of treated mice at day 12 post-treatment (Fig. 5e) and in both tumours and spleens, wild-type FOXO1-overexpressing CD8+ CAR T cells exhibited a less-differentiated phenotype (Fig. 5f,g). FOXO1 overexpression, but not TCF7 overexpression, also significantly reduced expression of the exhaustion markers PD-1 and TIM3 on tumour-infiltrating CD8+ CAR T cells, whereas Ki-67 expression was unchanged (Fig. 5h and Extended Data Fig. 13c,d). Enhanced infiltration of FOXO1-overexpressing CAR T cells was also observed at day 7 post-treatment, when control CAR T cells were almost absent from the tumour site (Extended Data Fig. 13e). This was despite the fact that the frequency of FOXO1-expressing CAR T cells was not increased in the spleen at this timepoint, indicating that FOXO1-expressing CAR T cells were more likely to traffic to the tumour site (Extended Data Fig. 13f). Analysis of serum from mice treated with control and FOXO1-expressing CAR T cells indicated no significant changes in the levels of enzymes associated with liver and kidney function (Extended Data Fig. 13g) or cytokines associated with cytokine release syndrome (Extended Data Fig. 13h). Moreover, treated mice elicited no significant changes in body mass during therapy, supporting the safety of this approach (Extended Data Fig. 13i). Finally,

we evaluated this approach in the context of T cells derived from the initial apheresis product derived from six patients with solid cancers who underwent Lewis Y CAR T cell therapy. FOXO1 overexpression resulted in an increased population of CD45RA+CD62L+ stem-like T cells and reduced expression of TIM3 and CD39 (Fig. 5i,j). Consistent with healthy donor derived CAR T cells, overexpression of FOXO1 in patient-derived CAR T cells led to a significantly enhanced maximal respiratory rate and spare respiratory capacity (Fig. 5k). To confirm that FOXO1 overexpression could modify patient CAR T cell phenotype and function in vivo, we treated OVCAR-3 tumour-bearing mice with CAR T cells derived from two of the patients. In both cases, FOXO1 expression in the patient-derived CAR T cells significantly enhanced the numbers of CAR T cells in both the spleens and tumours of treated mice (Fig. 5l). These CAR T cells also exhibited an increased proportion of CD45RA+CD62L+ stem-like phenotype in the spleen (Fig. 5m) and a less-exhausted phenotype in the tumours, characterized by reduced expression of PD-1 and TIM3 (Fig. 5n), indicating that FOXO1 overexpression can similarly modulate patient-derived CAR T cells towards a less-differentiated state.

## Discussion

CAR T cells are now established as an effective therapy for the treatment of a number of haematological malignancies in which patient responses are correlated with a less-differentiated phenotype and an ability to achieve long-term persistence[8,42]. A number of studies have explored mechanisms to enable CAR T cells to adopt less-differentiated phenotypes such as the use of IL-15, small molecule inhibitors such as AKT inhibitors[43], PI3K inhibitors (ClinicalTrials.gov ID NCT03274219), or epigenetic modifiers such as JQ1[20]. However, although small molecule inhibitors such as AKTi can be applied easily to CAR T cells in culture, the disadvantage for such approaches is that the effect is transient, and once CAR T cells are infused into the patient they differentiate and exhaust in a normal manner. In this study, we set out to

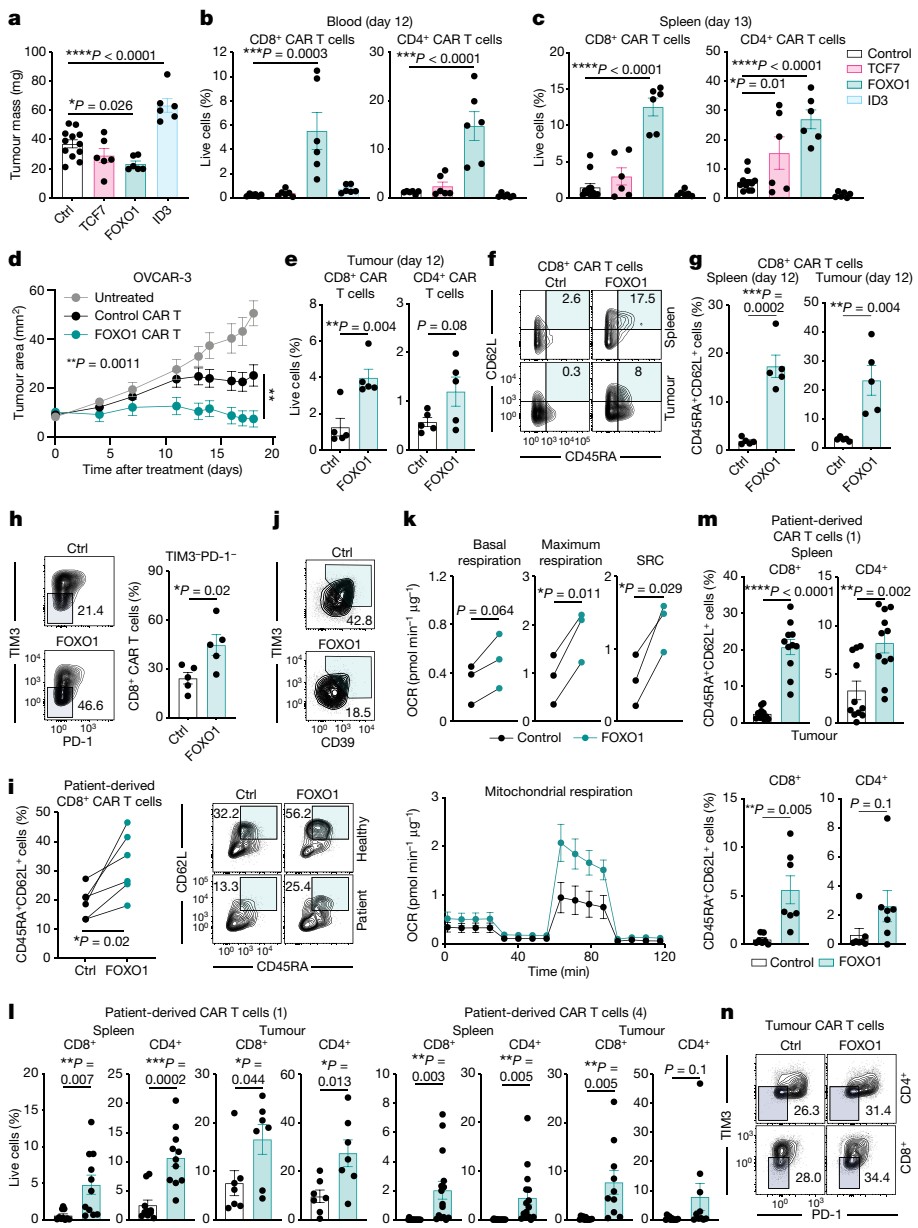

**Fig. 5 | FOXO1 overexpression enhances the efficacy of human CAR T cells in a mouse tumour transplant model. a–h**, Mice bearing OVCAR-3 tumours were treated with Lewis Y CAR T cells generated as in Fig. 3. **a**, Tumour mass at day 13 after treatment. **b,c**, Analysis of CAR T cell frequency in the blood (**b**) and spleens (**c**) of treated mice. **a–c**, Data are mean ± s.e.m. of *n* = 12 (control) or *n* = 6 (TCF7, FOXO1, ID3) (**a,c**), and *n* = 6 (**b**) mice per group. d. Therapeutic efficacy of CAR T cell treatment. Data are the mean ± s.e.m. of 7 mice per group from a representative experiment of *n* = 2. **e–h**, Analysis of CAR T cell frequency and phenotype in the spleens and tumours of treated mice at day 12 after treatment. Data are mean ± s.e.m. of 5 mice per group. **i**, Proportion of CD8⁺ CAR T cells generated from six patients enroled in a trial of CAR T cells with a CD45RA⁺CD62L⁺ phenotype. **j**, Representative FACS plot of TIM3 and CD39 expression on CD8⁺

CAR T cells derived from patients. **k**, Analysis of oxidative consumption in patient-derived CAR T cells by Seahorse MitoStress assay. Data represent paired analysis from three independent patients (top) or a representative patient (bottom). **l–n**, Analysis of CAR T cell frequency (**l**) and phenotype (**m,n**) in the spleens and tumours of mice treated with patient-derived CAR T cells at day 13 after treatment. **l**, Data are mean ± s.e.m. of *n* = 10 (patient 1, spleen control and patient 4, tumour FOXO1), *n* = 11 (patient 1, spleen FOXO1 and patient 4, tumour control), *n* = 7 (patient 1, tumour) or *n* = 15 (patient 4, spleen) mice per group pooled from two independent experiments. **m**, Data are mean ± s.e.m. of *n* = 11 (spleen) or *n* = 7 (tumour) mice per group pooled from two independent experiments. *P* values deterined by two-sided, one-way ANOVA (**a–c**), two-way ANOVA (**d**), two-sided unpaired *t*-test (**e,g,h,l,m**) or two-sided paired *t*-test (**i,k**).

identify transcriptional regulators that could genetically enforce the less-differentiated phenotype and improved metabolic fitness of IL-15 preconditioned CAR T cells. FOXO1 stood out as a major candidate of interest on the basis of the significant enrichment of FOXO1 target genes in CAR T cells cultured with IL-15 and its previously documented role in T cell memory formation[24] and metabolism[44]. We therefore selected FOXO1-ADA as a primary candidate, which was benchmarked against TCF7-, ID3- and JUN-overexpressing CAR T cells. Notably, FOXO1-ADA

was the most effective transcription factor for enhancing CAR T cell polyfunctionality. Furthermore, FOXO1-ADA overexpression significantly enhanced CAR T cell therapeutic activity in multiple syngeneic models, which was concomitant with improved metabolic fitness.

Overexpression of FOXO1-ADA, TCF7 or ID3 led to significantly increased CAR T cell numbers in the dLNs of treated mice. Notably, the number of FOXO1-ADA CAR T cells in the dLN correlated to the increased frequency of IFNγ- and TNF-positive intratumoral CAR T cells,

suggesting that lymph node residency may have a previously unappreciated role in maintaining CAR T cell polyfunctionality. This is of interest because of recent data indicating that the dLN is a key site for the maintenance of precursor exhausted T cells and response to immune checkpoint blockade[45]. Moreover, the dLNs were identified as a key site in the mechanism by which an mRNA vaccine strategy enhanced CAR T cell responses[46]. It therefore follows that a lack of lymph node residence observed for control CAR T cells may contribute to their dysfunction in the solid tumour setting and that engineering strategies that result in dLN residency may lead to improved CAR T functionality. Although lymph node residency may have contributed to the improved metabolic phenotype of FOXO1-ADA expressing CAR T cells in vivo, there are clearly cell autonomous effects of FOXO1 overexpression, since RNA-seq analysis of control and FOXO1-ADA-overexpressing CAR T cells in vitro revealed an enrichment of genes related to oxidative phosphorylation, and decreased expression of glycolytic enzymes prior to activation. This was further supported by Seahorse analysis of human FOXO1-overexpressing CAR T cells, which revealed a significantly enhanced basal and maximal respiratory capacity, consistent with our in vivo flow analyses of mouse CAR T cells expressing FOXO1-ADA.

In human CAR T cells, we observed that wild-type FOXO1 was the most effective strategy to enhance CAR T cell function, which may be owing to the enhanced level of transgene expression observed in human CAR T cells in our hands. Whereas expression of FOXO1-ADA prevented human CAR T cells from acquiring effector functionality such as cytokine production, overexpression of wild-type FOXO1 appeared to achieve a 'balanced' situation in which a less-differentiated phenotype can be maintained in the steady state but robust effector cell differentiation can still be achieved once the CAR becomes activated, as reflected by the fact that the majority of epigenetic and transcriptional changes enforced by FOXO1 were reversible upon stimulation through the CAR. In this regard, wild-type FOXO1 appears to exhibit favourable metabolic characteristics, given that FOXO1-expressing cells exhibit reduced expression of glycolysis-related genes in the steady state but can strongly upregulate these pathways upon activation. Overexpression of wild-type FOXO1 in human CAR T cells led to significantly enhanced tumour regression in mice. This was associated with increased tumour infiltration of FOXO1-expressing human CAR T cells and without overt signs of toxicity, supporting the safety and potential clinical applicability of this approach. Notably, in the context of human CAR T cells, we observed that FOXO1 was significantly more able to promote the emergence of CD45RA[+]CD62L[+] cells relative to TCF7 and ID3. Further comparison between FOXO1- and TCF7-overexpressing CAR T cells revealed that FOXO1 was significantly more able to drive transcriptional changes consistent with a more persistent and less-exhausted CAR T cell product. To further understand the mechanism underlying this effect, we performed scRNA-seq analysis, which indicated that FOXO1 expression promoted the emergence of a sub-population of CD8[+]CAR T cells characterized by high expression of *CD62L*, *IL7R* and *KLF2*. This population appeared largely responsible for the gene signatures of reduced exhaustion and glycolysis, and also notably expressed high levels of *CCR7*, suggesting that our observations of increased lymph node residency may also be reflected in human CAR T cells engineered to express FOXO1.

Although clinical trials assessing the use of IL-15 to generate less-differentiated CAR T cells are ongoing (ClinicalTrials.gov IDs NCT05359211, NCT05103631, NCT04715191, NCT04377932, NCT03721068 and NCT02992834), the effects of such cells are transient and do not provide long-term protection against exhaustion once CAR T cells become chronically stimulated in the solid tumour microenvironment. Our study presents a more durable approach to maintaining CAR T cell fitness and persistence while promoting polyfunctionality and anti-tumour efficacy without evidence of toxicity. Therefore, our study holds broad clinical potential for enhancing CAR T cell efficacy in solid cancers.

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

# Methods

## Ethical approval statement

This research and all study protocols have been approved and comply with the Peter MacCallum Animal Experimental Ethics Committee (AEC) ethical regulations regarding the use of animals. Studies using human peripheral blood mononuclear cells (PBMCs) from healthy donors was approved by the Peter MacCallum Cancer Centre Human Research Ethics committee. Informed consent was obtained from the Australian Red Cross.

## Animal models

C57BL/6 wild-type mice and C57BL/6 human-HER2 transgenic mice[47] were bred in the Peter MacCallum Cancer Centre animal facility. NOD. Cg-$Prkdc^{scid}$ $Il2rg^{tm1Wjl}$SzJ (NSG) mice were either bred at the Peter Mac-Callum Cancer Centre or obtained from Australian BioResources. Mice used in experiments were between 6 to 16 weeks of age and were housed in PC2 specific pathogen-free conditions and a minimum of 3 mice per group were used in each experiment. Mice were randomized prior to treatment according to tumour size to ensure all groups had equivalent tumour burden prior to therapy. Experiments were not blinded as the same investigators performed and analysed experiments and so blinding was not possible. Experiments were approved by the Animal Experimentation Ethics Committee no. E582, E671 and E693 and all experiments complied with the ethical endpoints stated in the approved projects, including maximum tumour size.

## Cell lines

The mouse MC38 colon adenocarcinoma cell line was provided by J. Schlom. The mouse breast carcinoma cell line E0771 was obtained from R. Anderson. The parental MC38 and E0771 tumour cell lines were retrovirally transduced with a mouse stem cell virus (MSCV) vector to express a truncated human HER2 antigen that lacks intracellular signalling components. Transduced tumour cell lines are referred to as MC38-HER2 and E0771-HER2. OVCAR-3 and MCF7 tumour cells were obtained from the American Type Culture Collection. PCR analysis was used to verify that tumour lines were negative for *Mycoplasma*.

Retroviral vector packaging cell lines PA317 and GP+E-86 were obtained from American Type Culture Collection (ATCC). The GP+E-86 and tumour cell lines were maintained in RPMI medium (Gibco Life Technologies) supplemented with 10% heat-inactivated fetal bovine serum, 1 mM sodium pyruvate, 2 mM glutamine, 0.1 mM non-essential amino acids, 10 mM 4-(2-hydroxyethyl)-1-piperazineethanesulfonic acid (HEPES), 100 U ml$^{-1}$ penicillin and 100 μg ml$^{-1}$ streptomycin. These cells were maintained at 37 °C in a humidified incubator with 5% $CO_2$. The PA317 cell line was maintained in DMEM (Gibco) supplemented with 2 mM glutamine and 100 U ml$^{-1}$ penicillin and 100 μg ml$^{-1}$ streptomycin and was maintained in a humidified incubator at 37 °C with 10% $CO_2$.

## Reagents and cytokines

Mouse IFNγ antibody (H22, IgG, BE0312) and the isotype control (2A3 clone, IgG2a, BE0254) antibody were purchased from BioXcell. The cytokine IL-2 was obtained from the National Institutes of Health and purchased from Peprotech. IL-7 and IL-15 were purchased from Peprotech. Where indicated, CAR T cells were stimulated with an anti-idiotype antibody that was custom made.

## Generation of retroviral packaging lines for the transduction of primary mouse T cells

cDNA encoding mouse TCF7, FOXO1 (wild-type), FOXO1-ADA, ID3 and JUN were cloned into the mouse stem cell virus (MSCV) vector encoding either an mCherry marker gene or truncated (lacks cell signalling components) human nerve growth factor receptor (NGFR). The viral packaging GP+E-86 cell line that produces the anti-HER2 CAR retrovirus was generated as previously described[48]. The anti-HER2 CAR construct was comprised of an extracellular scFv specific for human HER2, an extracellular CD8 hinge region, a CD28 transmembrane domain and an intracellular CD3ζ domain. GP+E-86 cell lines encoding both the anti-HER2 CAR and a transcriptional regulator were generated and the resulting anti-HER2 CAR packaging cells were sorted based on NGFR or mCherry expression by flow cytometry. Supernatants from these cells were used to transduce primary mouse T cells as previously described[12] and following transduction, CAR T cells were maintained in supplemented RPMI medium with IL-7 (200 pg ml$^{-1}$), IL-15 (10 ng ml$^{-1}$) and β-mercaptoethanol (50 μM).

## Generation of lentivirus for the transduction of human T cells

Lentiviral packaging plasmids (pCMV-VSV-G, pMDLg/pRRE, pRSV-Rev) and plasmid vectors encoding a second-generation Lewis Y CAR and either wild-type FOXO1, FOXO1-ADA or mCherry were purchased from GenScript. In brief, packaging plasmids and transgene plasmids were transfected into HEK293T cells. Across the following 3 days, cell culture supernatants were collected, pooled and centrifuged with Lenti-X-Concentrator (Takara Bio) to concentrate lentivirus. Lentivirus was used to transduce human T cells activated with OKT3 (30 ng ml$^{-1}$) and IL-2 (600 IU ml$^{-1}$) for 48 h by adding virus directly to cell cultures at a multiplicity of infection of 0.5 in Lentiboost (Sirion).

## CRISPR–Cas9 editing of CAR T cells

CRISPR–Cas9 editing of mouse CAR T cells was performed as described[12]. Per $20 \times 10^6$ naive splenocytes or $1 \times 10^6$ activated human PBMCs, 270 pmol single guide RNA (sgRNA) (Synthego) and 37 pmoles recombinant Cas9 were combined and incubated for 10 min to generate Cas9–sgRNA ribonuclear protein (RNP). Cells were resuspended in 20 μl P3 buffer (Lonza), combined with RNP and electroporated with a 4D-Nucleofector (Lonza) with pulse code E0115 or DN100 for human and mouse T cells, respectively. Prewarmed medium was then added to cells for 10 min prior to activation and transduction of mouse T cells or immediate transduction of human T cells. sgRNA sequences used were as follows: *Foxo1* guide 1: 5′-CACCUGGGGCGCUUCGGCCA-3′, guide 2: 5′-CCACUCGUAGAUCUGCGACA-3′. *FOXO1* guide 1 5′-CACC UGAGGCGCCUCGGCCA-3′.

## In vitro re-stimulation assay

Tumour cell targets were co-cultured with CAR T cells at a 1:1 ratio for 24 h. After overnight incubation, supernatants were collected and an equivalent number of tumour cells were reseeded into the incubations for another 24 h. This process was repeated one final time before cells were collected for analysis by flow cytometry and supernatants were analysed by cytometric bead array (CBA) using either mouse or human cytokine Flex sets (BD Biosciences) according to the manufacturer's instructions.

## Flow cytometry and cell sorting

For flow cytometric analysis Fc receptor block (2.4G2 diluted 1:50 from hybridoma supernatant in FACS buffer) was added to cells for 10 min at 4 °C. Cells were stained with 50 μl fluorochrome-conjugated antibody cocktails and incubated for 30 min in the dark at 4 °C. For intracellular staining, cells were fixed and permeabilized using the eBioscience FoxP3/Transcription Factor Staining Buffer Set (Thermo Fisher) according to the manufacturer's instructions. Samples were quantified using counting beads (Beckman Coulter; 20 μl per sample) using the following formula: number of beads per sample/bead events × cell events of interest. Cells were analysed on a BD LSRFortessa or BD FACSymphony (BD Biosciences) and data were analysed using Flowjo (TreeStar) or OMIQ (https://www.omiq.ai/). Cells were sorted using a BD FACSAria Fusion. See Supplementary Fig. 1 for the list of antibodies used for flow cytometry. Examples of the gating strategy used to identify mouse and human CAR T cells ex vivo is shown in Supplementary Fig. 1.

## Treatment of mice with CAR T cells

C57BL/6 human-HER2 transgenic mice were injected with $2 \times 10^5$ E0771-HER2 breast carcinoma cells orthotopically into the mammary fat pad 5–7 days prior to treatment or subcutaneously with $2.5 \times 10^5$ MC38-HER2 colon adenocarcinoma cells 5 days prior to treatment. After tumours were established, mice bearing E0771-HER2 or MC38-HER2 tumours were preconditioned with 4 Gy or 0.5 Gy total body irradiation respectively. Mice were then treated with intravenous doses of $1 \times 10^7$ CAR T cells on 2 consecutive days and one dose of IL-2 (50,000 IU per dose) with the first dose of CAR T cells, followed by two doses of IL-2 each day on the next 2 consecutive days. Tumour area was measured every 2–3 days following treatment. For IFNγ-blockade experiments, mice were dosed with 250 μg of anti-IFNγ or isotype control antibody 2A3 on days 0, 1 and 7 following CAR T cell treatment.

For experiments utilizing human anti-Lewis Y CAR T cells, NSG mice were injected with $5 \times 10^6$ OVCAR-3 tumour cells. Once tumours were established, at day 10–15 post injection, mice were treated with 1 Gy total body irradiation and intravenously treated with $2–5 \times 10^6$ Flag+ CAR T cells. Mice were treated with IL-2 as per experiments in the C57BL/6 human-HER2 transgenic model.

## Analysis of immune subsets in tumour, spleen, dLNs and blood

Blood was collected via submandibular or retroorbital bleed into tubes containing EDTA prior to euthanasia. Blood and spleen samples were treated twice or once respectively with ACK lysis buffer before staining for flow cytometry. Tumours were digested in SAFC DMEM medium (Gibco) with 0.01 mg ml$^{-1}$ DNase (Sigma Aldrich) and 1 mg ml$^{-1}$ type IV collagenase (Sigma Aldrich) for 30 min at 37 °C. Following digestion, tumour samples were filtered twice through a 70-μm filter to create a single-cell suspension and resuspended in Fc block prior to staining for analysis by flow cytometry. For stimulation of intratumoral CAR T cells to assess cytokine secretion capacity, tumour cell suspensions were resuspended in complete RPMI medium with 10 ng ml$^{-1}$ phorbol 12-myristate 13-acetate (Abcam), 1 μg ml$^{-1}$ ionomycin (Abcam), GolgiStop (1:1,500 dilution, BD Biosciences) and GolgiPlug (1:1,000 dilution, BD Biosciences). Samples were incubated for 3 h at 37 °C with 5% $CO_2$ prior to staining for analysis by flow cytometry. Single-cell suspensions from dLN were created by placing tissue between two pieces of 70-μm filter mesh in 400 μl of FACS buffer and by mechanically digesting using the end of a syringe. The resultant cell suspension was then stained for analysis by flow cytometry. For mitochondrial analysis, isolated cells were stained using Mitotracker Deep Red FM and Mitotracker Green FM (Thermo Fisher) according to the manufacturer's protocols.

## Seahorse assay

A Seahorse XFe24 Bioanalyser (Agilent) was used to determine OCR for indicated CAR T cells prepared from 5 separate donors. Cells were washed in assay medium (XF Base media (Agilent) with glucose (10 mM), sodium pyruvate (1 mM) and L-glutamine (2 mM) (Gibco), pH 7.4 at 37 °C) before being plated onto Seahorse cell culture plates coated with Cell-Tak (Corning) at $4 \times 10^5$ cells per well. After adherence and equilibration, cellular OCR and extracellular acidification rates (ECAR) were measured using a Seahorse MitoStress assay (Agilent), with addition of oligomycin (1 μM), carbonyl cyanide 4-(trifluoromethoxy) phenylhydrazone (1.2 μM) and antimycin A and rotenone (0.5 μM each). Assay parameters were as follows: 3 min mix, no wait, 3 min measurement, repeated 3 times at basal and after each addition. Raw OCR values were normalized to the amount of protein per well, as assessed by a Pierce BCA protein assay (Thermo Fisher) performed as per manufacturer instructions. SRC was calculated as OCR at maximum rate − OCR in basal state.

## Gene expression analysis

Following manufacturer's instructions, RNA-seq libraries were prepared from RNA using the Quant-seq 3′ mRNA-seq Library Prep Kit for Illumina (Lexogen). Single-end, 75 bp RNA-seq was performed via NextSeq (Illumina) and CASAVA 1.8.2 was subsequently used for base calling. Cutadapt v2.1 was used to remove random primer bias and trim 3′ end poly-A-tail derived reads. Quality control was assessed using FastQC v0.11.6 and RNA-SeQC v1.1.8[49]. Sequence alignment against the mouse reference genome mm10 or the human genome hg19 was performed using HISAT2. Finally, featureCounts from the Rsubread software package 2.10.5 was used to quantify the raw reads with genes defined from the respective Ensembl releases[50]. Gene counts were normalized using the TMM (trimmed means of M-values) method and converted into log$_2$-transformed counts per million (CPM) using the EdgeR package[51,52]. The quasi-likelihood $F$-test statistical test method based on the generalized linear model (glm) framework from EdgeR was used for differential gene-expression comparisons adjusted $P$ values were computed using the Benjamini–Hochberg method. Principal component analysis was performed generated based on the top most variable genes. DEGs were classified as significant based on a false discovery rate cut-off of less than 0.05. For heat maps, the pheatmap R package was used to plot row mean centred and scaled normalized log$_2$(CPM + 0.5) values. Genes columns or rows were sorted by hierarchical clustering using Euclidean distance and average linkage.

Unbiased gene set enrichment analysis was performed using fgsea package on differential expressed genes pre-ranked by fold change with 1,000 permutations (nominal $P$ value cut-off <0.05)[53]. Reference gene sets were obtained from the MsigDB library for Hallmarks, KEGG (https://www.genome.jp/kegg/kegg1.html), CHEA dataset[54–57], or based on previously published analyses of glycolysis signature[23], scRNA-seq-derived T cell clusters in patients[11].

## scRNA-seq data processing and analysis

CAR T cells were co-cultured with MCF7 tumour cells at a 1:1 ratio for 24 h. Fc-receptors were blocked with human Fc Block (BD Biosciences) for 10 min at 4 °C before staining with 50 μl fluorochrome-conjugated antibody cocktail for 30 min in the dark at 4 °C. Samples were labelled with anchor lipid-modified oligo (LMO) (5′-TGGAAT TCTCGGGTGCCAAGGgtaacgatccagctgtcact-[lipid]-3), co-anchor LMO (5′-[lipid]-AGTGACAGCTGGATCGTTAC-3′) and sample specific barcodes for 5 min in the dark at 4 °C. CAR+ T cells were sorted by FACS and samples were pooled at equal ratios followed by staining with 100 μl TotalSeq-C anti-human CD4 and CD8 (BioLegend) antibody cocktail for 30 min in the dark at 4 °C. scRNA-seq data were generated using the 10x Cell Ranger pipeline (7.1.0) and hg38 genome. Specifically, cellranger multi was used to generate raw feature barcode matrices. Downstream analysis was performed in R (version 4.2.0). Empty droplets were detected and removed from the raw feature barcode matrix using the emptyDrops function from the DropletUtils (version 1.16.0) package and doublets were detected and removed using DoubletFinder (verison 2.0.3). Using Seurat (version 4.3.0), cells with less than 200 features and more than 5% mitochondrial reads were excluded. Standard Seurat data processing and normalization steps were performed: NormalizeData, FindVariableFeatures, ScaleData, RunPCA, RunUMAP, FindNeighbors and FindClusters; clusters with low-quality metrics were removed, and the final resolution was determined using results from the clustree package (version 0.5.0). LMOs were demultiplexed using HTODemux (Seurat). DEGs were calculated using the functions FindAllMarkers (Seurat) using a log$_2$-transformed fold change threshold of 0.125 and an adjusted $P$ value of less than 0.05, and included the number of counts as a latent variable. Pseudobulk DEGs were detected using the Libra package (version 1.0.0) using the run_de function. Gene-set enrichment was performed using the fgsea package with all expressed genes as the background gene list, which was ranked by average log-transformed fold change detected with FindMarkers using a log$_2$-transformed fold change threshold of 0 and min.pct parameter set to 0. To perform diffexp analyses and GSEA between individual

groups within each cluster, the to_psuedobulk function from Libra was used to pull out pseudobulk count matrix of each replicate pool and clusters. EdgeR and fgsea was then utilized to perform differential expression and gsea analyses of reference gene signatures. The single-cell signature explorer program was utilized for visualization of gene signatures across UMAP plots[58].

## ATAC-seq data analysis

Sequencing files for ATAC-seq experiments were demultiplexed using Bcl2fastq (v2.20) to generate Fastq files. Next, quality control of files were performed using FASTQC (v0.11.5). Adaptor trimming of paired-end reads was performed with NGmerge (v0.3) where required[59]. Alignment of reads to either the reference human (hg38) or mouse (mm10) genome was performed using Bowtie2 (v2.3.3). The resulting SAM files were converted to BAM files using Samtools (v1.4.1) using the view command, which were subsequently sorted and indexed, with potential PCR duplicates marked with Samtools markdup. Peak calling was performed with either MACS2 (v2.1.1) or Genrich (v0.6.0) packages. Annotation of ATAC-seq peaks to proximal genes was performed using either annotatePeaks.pl (Homer, v4.11) or the annotatePeak function from ChIPseeker R package (v1.8.6). BAM files were converted into Big-Wig files using the bamCoverage function (Deeptools, v3.5.0). BigWig files were then imported into Integrative Genomics Viewer (IGV, v2.7.0) for visualization of specific loci. To generate IGV style track plots from BigWig files, the package trackplot was used[60]. The HOMER make-TagDirectory command was used to generate tag directories, and the findPeaks command was used to identify peaks, with the control tag directory set to respective control groups. Motif discovery using the findMotifsGenome tool and default settings identified de novo motifs from peaks identified. The ChromVAR R package[61] was used to identify enriched motifs from the JASPAR 2022 database[62], in unstimulated or stimulated groups.

## Statistical analysis

Statistical analyses were performed using GraphPad Prism. Analyses performed include paired or unpaired Student's $t$-test to compare two datasets, one-way ANOVA to analyse multiple datasets across a single timepoint and two-way ANOVA when analysing multiple sets of data across time.

## Reporting summary

Further information on research design is available in the Nature Portfolio Reporting Summary linked to this article.

## Data availability

The RNA-seq and ATAC-seq data that support the findings of this study have been deposited in the NCBI Gene Expression Omnibus (GEO) under accession GSE225527 that contains the subseries GSE225521, GSE225522, GSE225523, GSE225526, GSE263162, GSE263256 and GSE263257. Reference gene sets were obtained from the MsigDB library for Hallmarks (https://www.gsea-msigdb.org/gsea/msigdb/human/collections.jsp), KEGG (https://www.genome.jp/kegg/kegg1.html), CHEA dataset[54–57], or based on previously published analyses of glycolysis signature[23] or scRNA-seq-derived T cell clusters in patients[11]. Reference databases used are hg38 (https://www.ncbi.nlm.nih.gov/datasets/genome/GCF_000001405.26/), mm10 (Mus musculus genome assembly GRCm38 - NCBI - NLM (nih.gov)) and hg19 (https://www.ncbi.nlm.nih.gov/datasets/genome/GCF_000001405.13/). Additional data are available upon request from the authors. Source data are provided with this paper.

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

**Acknowledgements** This work was funded by a programme grant and ideas grant from the National Health and Medical Research Council (NHMRC programme grant number 1132373, ideas grant 2012475 and a project grant from the National Breast Cancer Foundation (IIRS-22-095). P.A.B. was supported by a National Breast Cancer Foundation Fellowship (ECF-17-005, 2017- 2020) and a Victorian Cancer Agency Mid-Career Fellowship (MCRF20011, 2021–current). I.G.H. was supported by a Victorian Cancer Agency Early Career Fellowship (ECRF20017). P.K.D. was supported by NHMRC Senior Research Fellowships (APP1136680 (2018–2022) and 2026403 (2024–current)). J. Lai was a recipient of a US Cancer Research Institute Irvington postdoctoral fellowship (award no. 3530). I.A.P. was supported by a Victorian Cancer Agency Mid-Career Fellowship (2022- Current). A.M.S. was supported by an NHMRC Investigator Fellowship (1177837). C.Y.S. was supported by a mid-career Fellowship from the Victorian Cancer Agency (MCRF22022). The authors acknowledge the contributions of K. Gill, M. Rear and G. Sissing who act as consumer representatives for the laboratory, the Peter MacCallum Molecular Genomics, Flow Cytometry, Genotyping and Animal Facility Cores for their respective contributions to the study. The authors thank the Centre of Excellence in Cellular Immunotherapy at the Peter MacCallum Cancer Centre for supply of the lentiviral plasmid. Images in Fig. 1a and Extended Data Fig. 3a were created with BioRender.com.

**Author contributions** J.D.C., C.M.S., I.M., J. Lai, P.A.B. and P.K.D. designed the experiments, developed the methodology, analysed and interpreted data and wrote the manuscript. J.D.C., C.M.S., I.M., Jasmine Li, K.S., J.N.L., Jiawen Li, Y.-K.H., K.M.Y., A.X.Y.C., C.W.C., E.B.D, K.L.T., J.T., P.A.D., T.X.H., M.N.d.M., E.V.P., J.S.K., D.N., P.S.K.L., J.S., C.D., J.Z., I.G.H., K.M.Q., J. Lai, P.A.B. and P.K.D. performed experiments and acquired data. N.Y.L.S. analysed data. L.M.K., A.M.S., B.J.S., S.J.H., J.O., I.A.P., K.M.Q., P.J.N. and C.Y.S., provided technical support and advice on data analysis and interpretation. J. Lai, P.A.B. and P.K.D. supervised the study and were responsible for coordination and strategy.

**Competing interests** P.A.B. declares research funding from AstraZeneca, Bristol-Myers-Squibb and Gilead Sciences. P.K.D. declares research funding from Myeloid Therapeutics, Prescient Therapeutics, Bristol-Myers-Squibb and Juno Therapeutics. J. Lai is a present employee at oNKo-Innate. S.J.H. declares consultancy fees and honoraria from Celgene, Janssen Cilag and Novartis; declares research funding from Celgene, Janssen Cilag, Novartis and Haemalogix; was an investigator on studies for Celgene, Janssen Cilag, Novartis and Haemalogix; and has served on the advisory board for Celgene, Janssen Cilag, Novartis and Haemalogix. I.A.P. declares research funding from Bristol-Myers-Squibb and Astrazeneca. The other authors declare no competing interests.

**Additional information**
**Correspondence and requests for materials** should be addressed to Junyun Lai, Paul A. Beavis or Phillip K. Darcy.

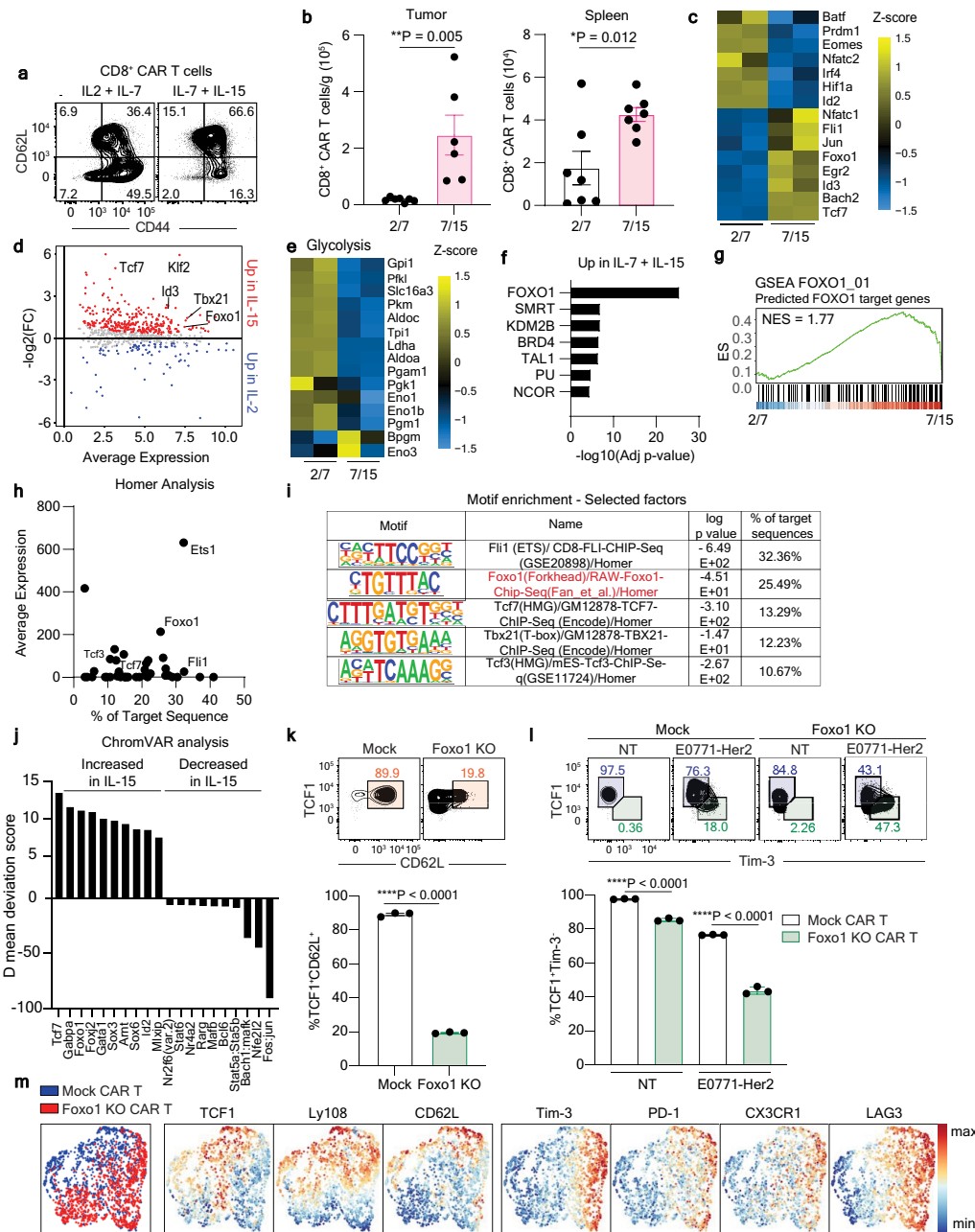

**Extended Data Fig. 1 | CAR T cells polarized with IL-15 have increased expression of Foxo1. a-j**, Murine anti-Her2 CAR T cells preconditioned with IL-2/IL-7 or IL-15/IL-7. **a**, Expression of CD62L and CD44 on CD8⁺ CAR T cells on concatenated samples from a representative experiment of n = >3. **b**, CAR T cell numbers from tumors and spleens of E0771-Her2 tumor bearing mice at day 9 post treatment. n = 7 (Tumor 2/7, spleen 2/7 and 7/15) or n = 6 (tumor 7/15) mice per group. **c-e**, Heatmaps and MA-plot of indicated genes in CD8⁺CD62L⁺ CAR T cells at day 6 post transduction. **f-g**. Gene set enrichment analyses of CAR T cells from **c** relative to the CHEA dataset (**f**) and in silico predicted FOXO1 target genes (**g**; FOXO_01; https://www.gsea-msigdb.org/gsea/msigdb/cards/FOXO1_01.html). **h-j**, CD8⁺ CAR T cells treated as per **c**. were analyzed by ATAC-Seq. **h-i**, Homer analysis of motif enrichment in differentially accessible peaks. The top 50 most significantly enriched motifs in IL-15 conditioned CAR

T cells plotted relative to the percentage of target sequence and average expression in IL-15 cultured CD8⁺CD62L⁺ CAR T cells. **j**, ChromVar analysis from **h**. Mean deviation score was calculated for the following subsets of CD8⁺ CAR T cells; CD62L⁺CD44⁺, CD62L⁺CD44^low, CD62L⁻CD44⁺. The delta mean deviation score for the CD62L⁺CD44⁺ subset is shown for the top 10 ranking transcription factors in each direction. **c-j**, Samples indicative of biological duplicates. **k-l**, anti-Her2 CAR T cells were CRISPR/Cas9-edited to target *Foxo1*. At day 5 post transduction CD8⁺ CAR T cells were phenotyped in steady state (**k**) or serially cocultured with E0771-Her2 tumor cells for 3 consecutive days (**l-m**). **k-m**, representative of >3 independent experiments, data points indicate biological triplicates. **b** Bars represent mean ± SEM **k-l**, Bars represent mean ± SD, ****p < 0.0001, unpaired two-sided *t* test. Source data are provided in the Source Data file.

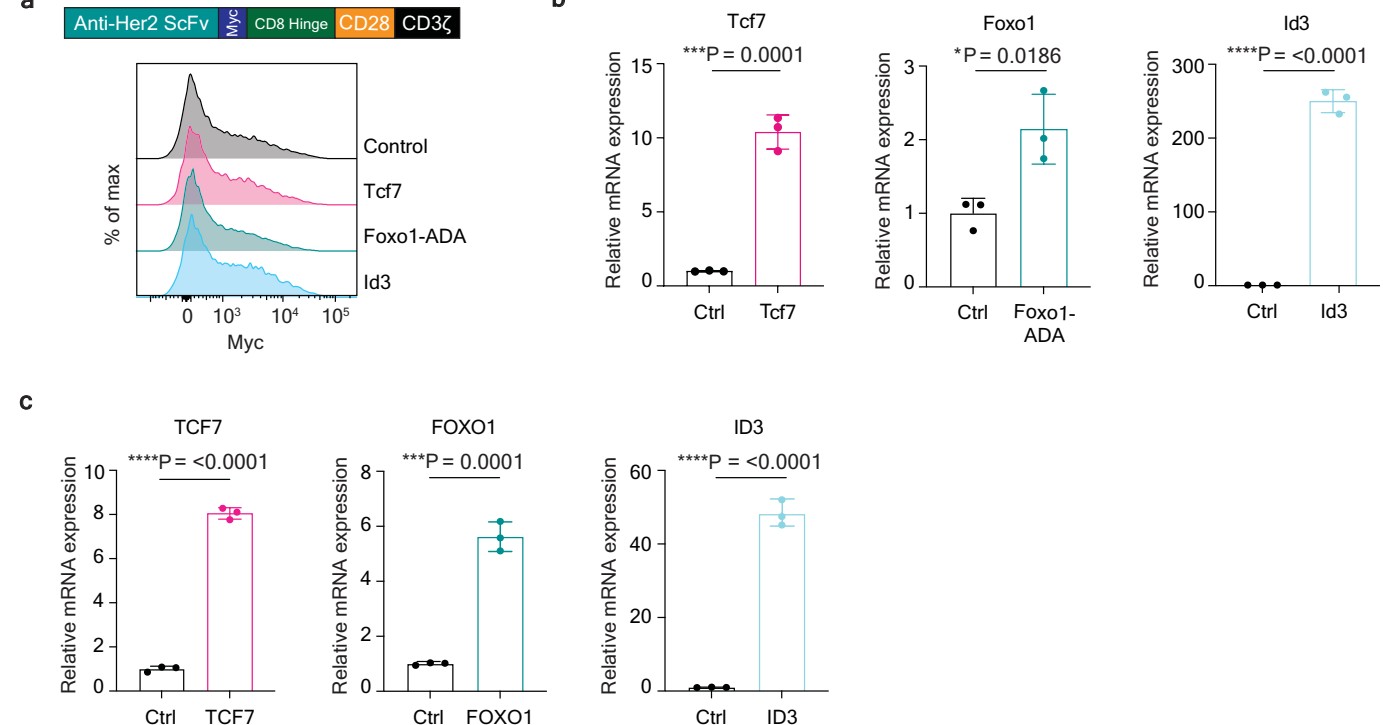

**Extended Data Fig. 2 | CAR T cell transduction efficiency and transgene overexpression. a**, Schematic of anti-Her2 CAR with Myc binding domain and detection of Myc tag in CAR T cells transduced with the indicated transcription factors. Representative histograms of 3 independent transductions. **b**, Expression of indicated transcription factors in murine CD8+NGFR+ CAR T cells (**b**) or human FACS sorted CAR T cells as determined by qRT-PCR. Data represent the mean ± SD of technical triplicates from two independent experiments. Statistics determined by unpaired two-sided t test, (*p < 0.05, ***p < 0.001, ****p < 0.0001). Source data are provided in the Source Data file.

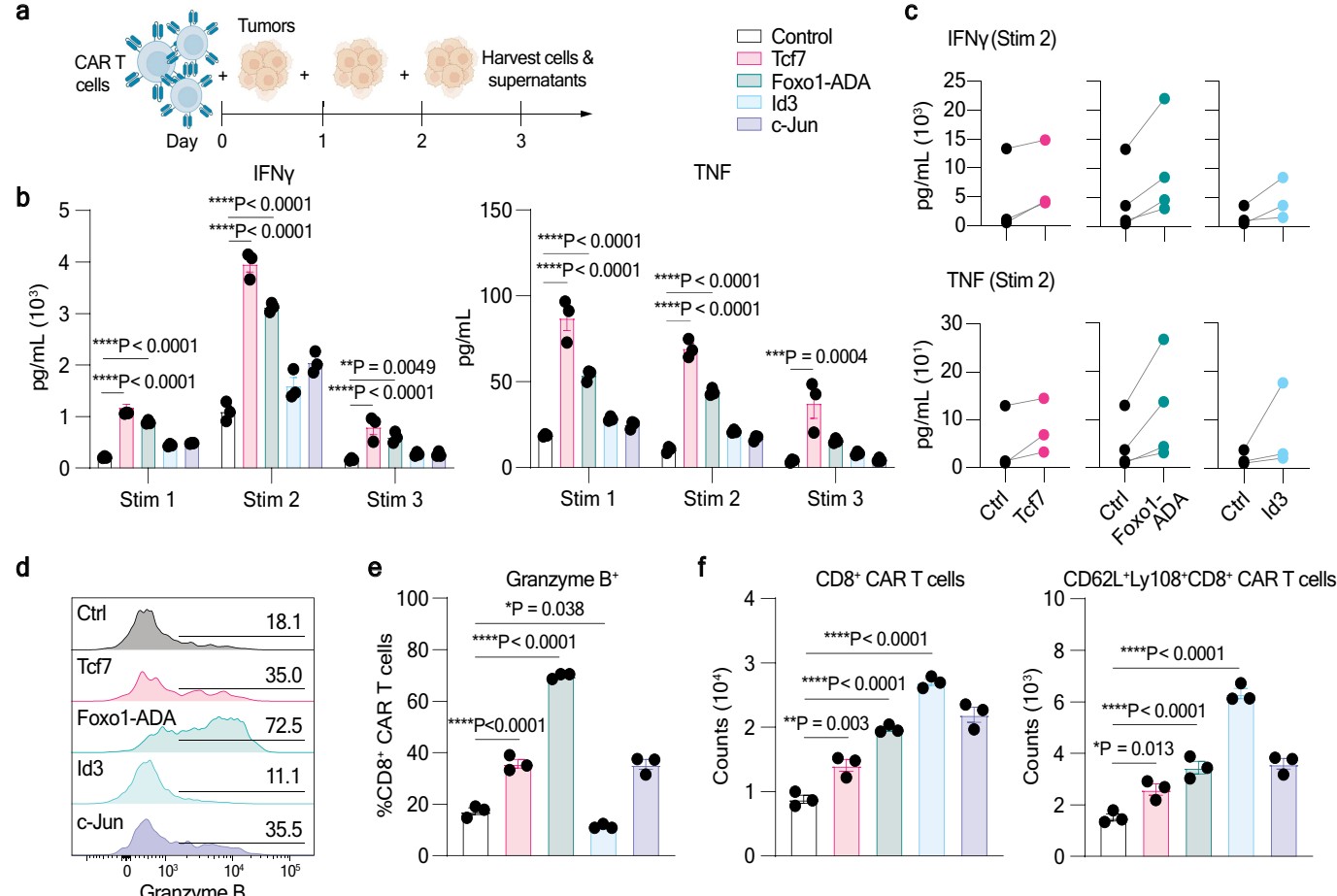

**Extended Data Fig. 3 | FOX1-ADA overexpression enhances CAR T cell polyfunctionality. a**, Schematic of tumor cell coculture assay. **b**, IFNγ and TNF production following each round of E0771-Her2 tumor cell stimulation with CAR T cells modified via overexpression of indicated transcription factors. **c**, Paired analyses of n = 3 (Tcf7, ID3) or n = 4 (Foxo1-ADA) repeat experiments setup as per **b**. **d-e**, Expression of Granzyme B in CAR T cells following 72 h of coculture. Histogram overlays of concatenated data from biological replicates. **f**, Number of total CD8+ or CD62L+Ly108+CD8+ CAR T cells following 72 h of coculture. **b**, and **e-f** Data represents the mean ± SD of technical triplicates. Statistics determined by two-sided, one-way ANOVA. Representative of at least 3 independent experiments. Source data are provided in the source data file.

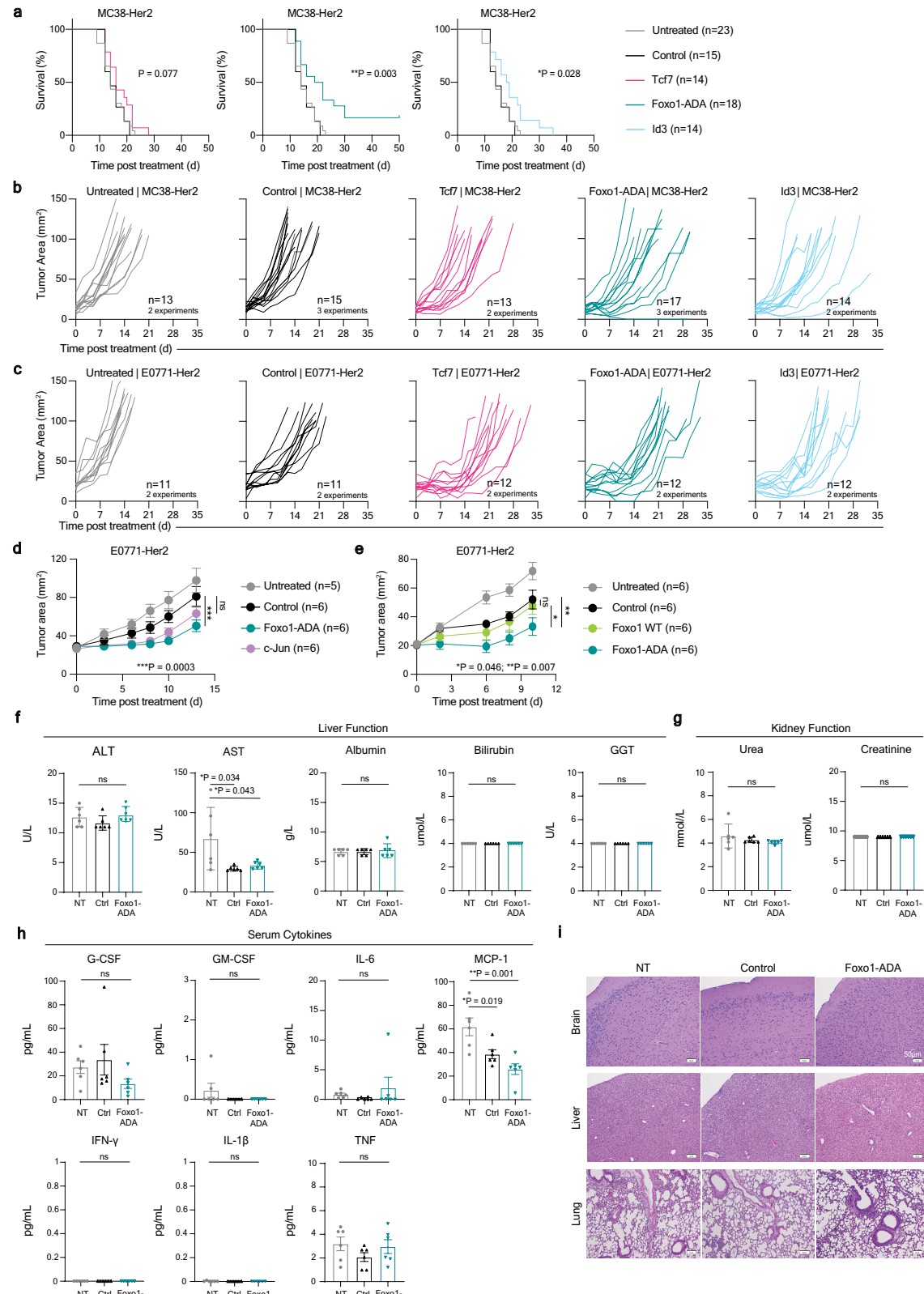

**Extended Data Fig. 4** | See next page for caption.

**Extended Data Fig. 4 | Therapeutic efficacy and safety of murine CAR T cells in E0771-Her2 and MC38-Her2 tumor bearing mice. a-b**, MC38-Her2 subcutaneous tumors or **c**, E0771-Her2 mammary fat pad tumors were established in mice for 5 to 7 days, prior to treatment with control, Tcf7, Foxo1-ADA or ID3 overexpressing CAR T cells as per Fig. 1. **a**, Survival curves of mice. Data represent 2 independent experiments (Tcf7, ID3) or 3 independent experiments (Untreated, Control, Foxo1-ADA) combined. Statistics determined by Mantel-Cox test, *, p < 0.05 **, p < 0.01. **b-c**, Individual tumor growth curves of mice. **d-e**, Therapeutic efficacy of control, wild-type Foxo1 (Foxo1 WT) overexpressing, Foxo1-ADA overexpressing, c-Jun overexpressing, or no CAR T cells. Data presented as mean ± SEM from indicated number of mice. Statistics determined by Two-way ANOVA (ns = not significant, *p < 0.05, **p < 0.01, ***p < 0.001). **f-h**, At day 9 post treatment, serum was taken from treated mice. Liver and kidney function was assessed by levels of indicated factors and potential cytokine release syndrome assessed through measurement of indicated cytokines. **f-h** Data represent the mean ± SD (**f, g**) or ± SEM (**h**) of 6 samples obtained from independent mice for each group. Statistics determined by One-way ANOVA (ns = not significant, *p < 0.05, **p < 0.01, ***p < 0.001). **i**, hematoxylin and eosin histology staining was performed on brain liver and lungs of mice day 9 post treatment. Images are representatives of 3 mice per group of one experiment. Source data are provided in the Source Data file.

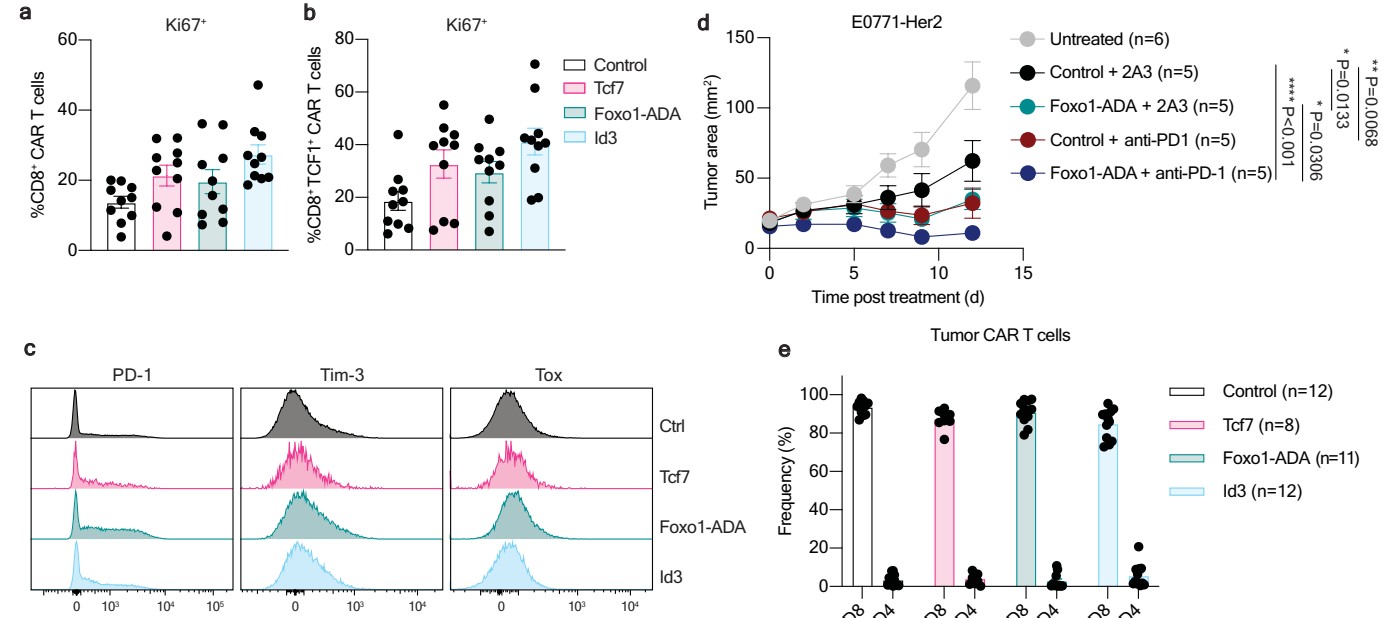

**Extended Data Fig. 5 | Phenotype of anti-Her2 CAR T cells isolated from E0771-Her2 expressing tumors.** E0771-Her2 tumor bearing mice were treated with anti-Her2 CAR T cells and tumors analyzed by flow cytometry at day 9 post treatment. **a**, Proportion of CD8+ **b**, or CD8+Tcf1+ CAR T cells expressing Ki67. Data presented as mean ± SEM n = 10 mice per group combined from 2 experiments. **c**, Expression of PD-1, TIM-3 and Tox in CD8+ CAR T cells modified with the indicated transcription factor. Data obtained from concatenated samples of n = 5 mice from a representative experiment of n = 2. **d**, E0771-Her2 tumor-bearing mice were treated with CAR T cells and with a total of 4 doses of 200 μg of anti-PD-1 or 2A3 on days 0, 3, 7 and 11 post-treatments. Data presented as mean ± SEM of indicated number of mice. Statistics determined by Two-way ANOVA (*p < 0.05, **p < 0.01, ****p < 0.0001). **e**, Frequency of CD8+ and CD4+ T cells within the NGFR+ (CAR+ subset). Data represented as the mean ± SEM of n = 12 (control and id3), n = 8 (Tcf7), n = 11 (Foxo1-Ada) mice combined from two independent experiments. Source data are provided in the Source Data file.

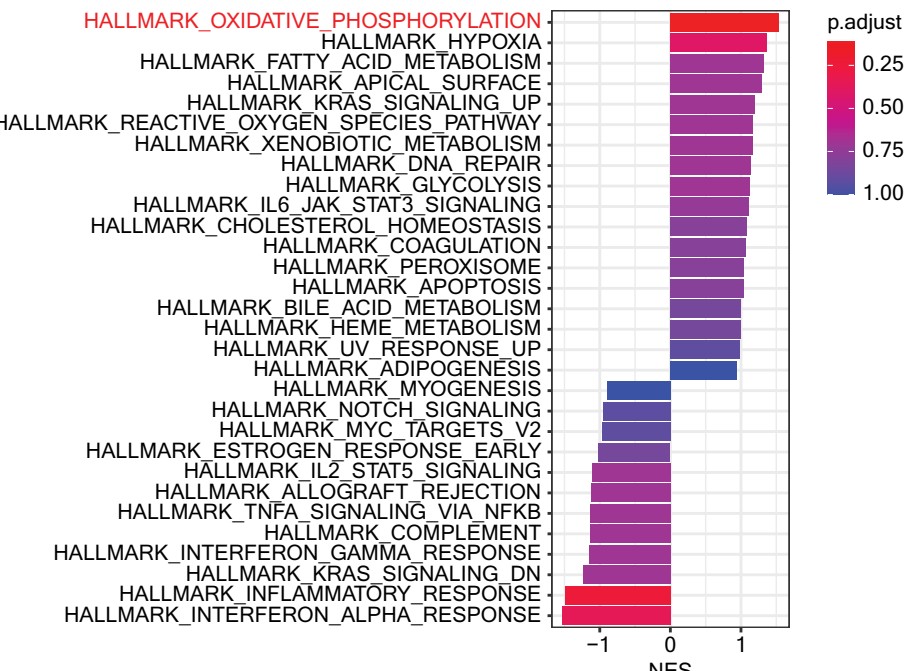

**Extended Data Fig. 6 | GSEA pathways in in vitro stimulated Foxo1-ADA overexpressing CAR T cells.** Foxo1-ADA and control CAR T cells were stimulated for 72 h with an agonistic antibody against the Her2 directed CAR prior to RNA-sequencing. Unbiased ranking of gene sets from the Hallmark gene sets.

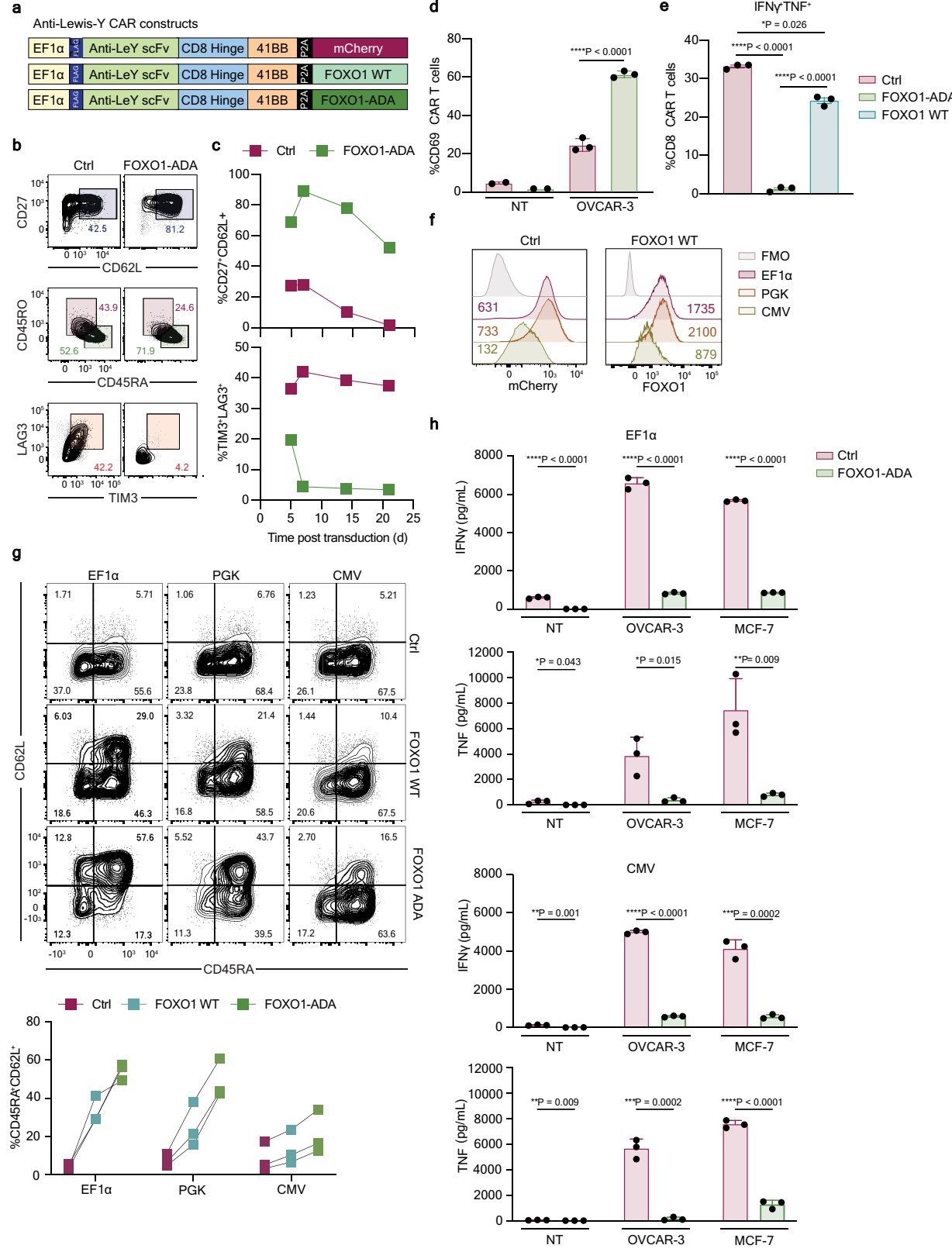

**Extended Data Fig. 7** | See next page for caption.

**Extended Data Fig. 7 | Comparison of FOXO1 WT and FOXO1-ADA in human CAR T cells. a**, LeY-mCherry (Control), LeY-FOXO1-ADA and LeY-FOXO1 WT constructs. **b**, Expression of indicated markers on healthy donor derived control or FOXO1-ADA CD8⁺ CAR T cells 5 days post transduction. Representative of 3 independent donors. **c**, Timecourse of control or FOXO1-ADA CD8⁺ CAR T cell phenotypes during in vitro culture. Data representative of 3 independent donors. **d-e**. CAR T cells were serially cocultured with OVCAR-3 tumor cells through 3 successive rounds of tumor cell addition. **d**, Proportion of CD8⁺ CAR T cells expressing CD69 in control or FOXO1-ADA expressing CAR T cells at Day 3. Data presented as mean ± SD of three technical replicates from a representative donor of n = 3. Statistics determined by unpaired two-sided t test. **e**, Proportion of CD8⁺ CAR T cells expressing IFNγ and TNF in control, FOXO1-ADA or FOXO1 WT expressing CAR T cells. Data presented as mean ± SD of three technical replicates from a representative donor of n = 3. Statistics determined by One-way ANOVA. **f**, T cells were transduced with the anti-Lewis Y CAR and mCherry or FOXO1 WT transgenes driven by the EF1α, PGK or CMV promoters. Histograms show the expression of mCherry (left) or FOXO1 (right). Numbers indicate MFI for relevant transgenes. Data representative of 3 independent donors. **g**, Flow cytometry analysis of the CD45RA⁺CD62L⁺ profile of CD8⁺ CAR T cells transduced with either mCherry (Ctrl), FOXO1 WT or FOXO1-ADA driven by the indicated promoters. Representative plots and paired data are shown for 3 independent donors. **h**. CAR T cells were cocultured for 24 h with OVCAR-3 or MCF7 tumor cells and production of IFNγ and TNF determined. Data is represented as the mean ± SD of experimental triplicates. Statistics determined by unpaired two-sided t test. *p < 0.05, **p < 0.01, ***p < 0.001, ****p < 0.0001. Source data are provided in the Source Data file.

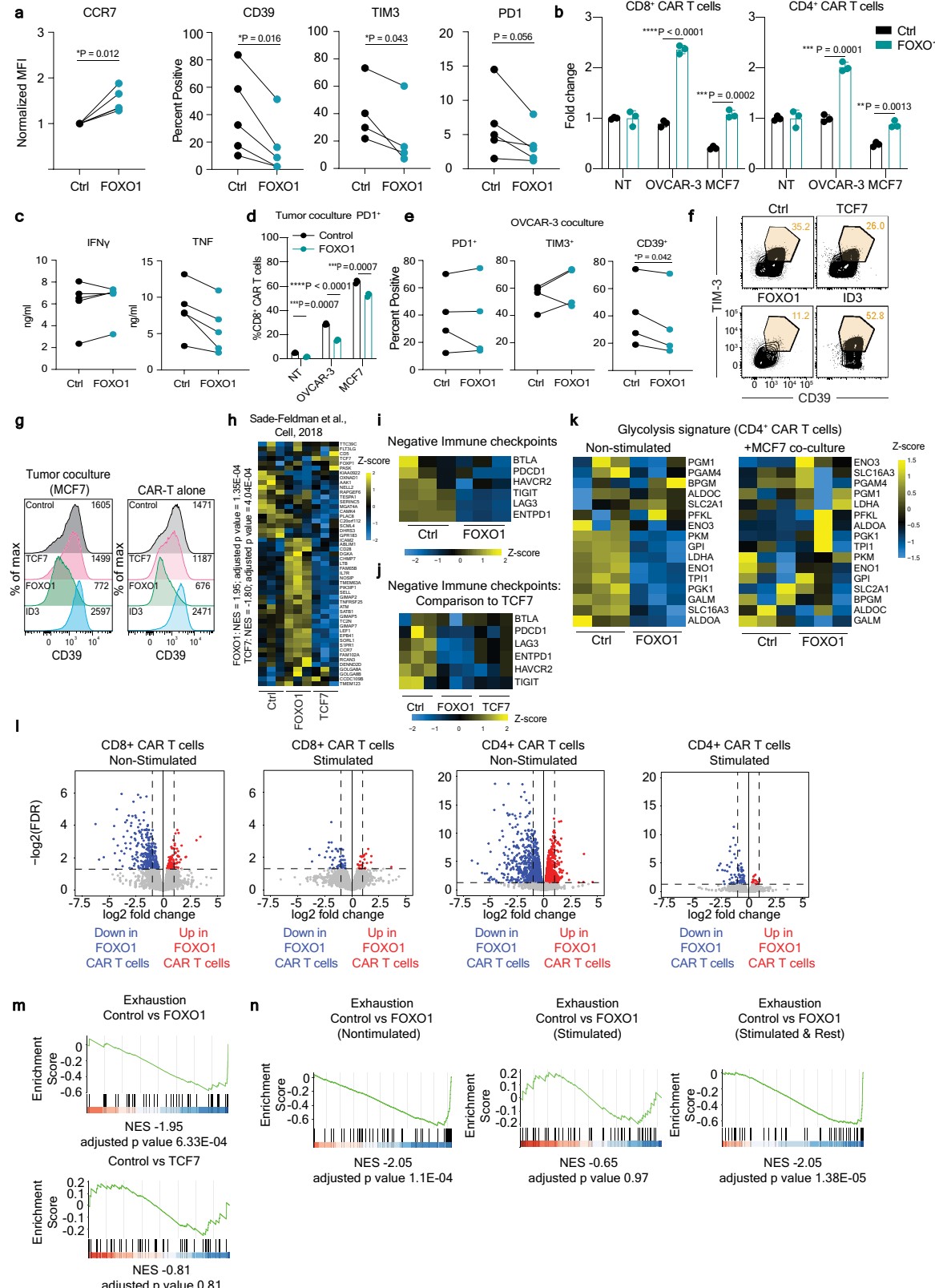

**Extended Data Fig. 8 | Phenotype and transcriptional profile of healthy donor CAR T cells following FOXO1 overexpression.** Healthy donor T cells were transduced with mCherry (ctrl), FOXO1, TCF7 or ID3 and an anti-Lewis Y CAR. **a**, Paired data of indicated phenotypic markers in control and FOXO1 expressing CAR T cells, n = 5 (CCR7, CD39, PD1) or 4 (TIM3) independent donors. Statistics determined by paired two-sided t test,. **b-e, g**, CAR T cells were serially cocultured with OVCAR-3 or MCF7 tumor cells over 72 h after which point CAR T cell number (**b**), cytokine production (**c**) or cell surface phenotype (**d-e, g**) were determined. **b**, **d**, Data represents the mean ± SD of biological triplicates from a representative experiment of n = 4. Statistics determined by unpaired two-sided t test. **c**, **e**, Paired data of indicated markers in control and FOXO1 expressing CAR T cells, n = 5 independent donors (**c**) or n = 4 independent donors (**e**). **e**, Statistics determined by paired two-sided t test. **f, g**, Flow cytometry showing the expression of CD39 and TIM3 prior to (**f, g**) or after co-culture with MCF7 tumor cells (**g**), representative of 2 independent experiments. **h-n**. Gene expression of CD8[+] CAR T cells (**h-j, l-n**) and CD4[+] CAR T cells (**k-l**) was determined by RNA-Sequencing as per Fig. 3. Data indicative of biological triplicates. **h**, enrichment of genes associated with less differentiated T cells that correlate with improved responses to immune checkpoint blockade in FOXO1 but not TCF7 expressing CAR T cells. **i-j**, Expression of indicated immune checkpoints in control, TCF7 or FOXO1 expressing CD8[+] CAR T cells **k**, Expression of genes associated with glycolysis. **l**, Volcano plot highlighting differentially expressed genes in control or FOXO1-expressing CAR T cells prior to and after tumor cell coculture. **m**, negative enrichment of genes associated with exhaustion in FOXO1-expressing but not TCF7-expressing CD8[+] CAR T cells, gene set defined by[39]. **n**, Control or FOXO1-expressing CD8[+] CAR T cells were assessed by RNA-Sequencing prior to stimulation (left), after 24 h activation with 0.1 µg/ml plate bound anti-Lewis Y (middle) or after 7 days recovery post stimulation with anti-Lewis Y (right). *p < 0.05, **p < 0.01, ***p < 0.001, ****p < 0.0001. Source data are provided in the Source Data file.

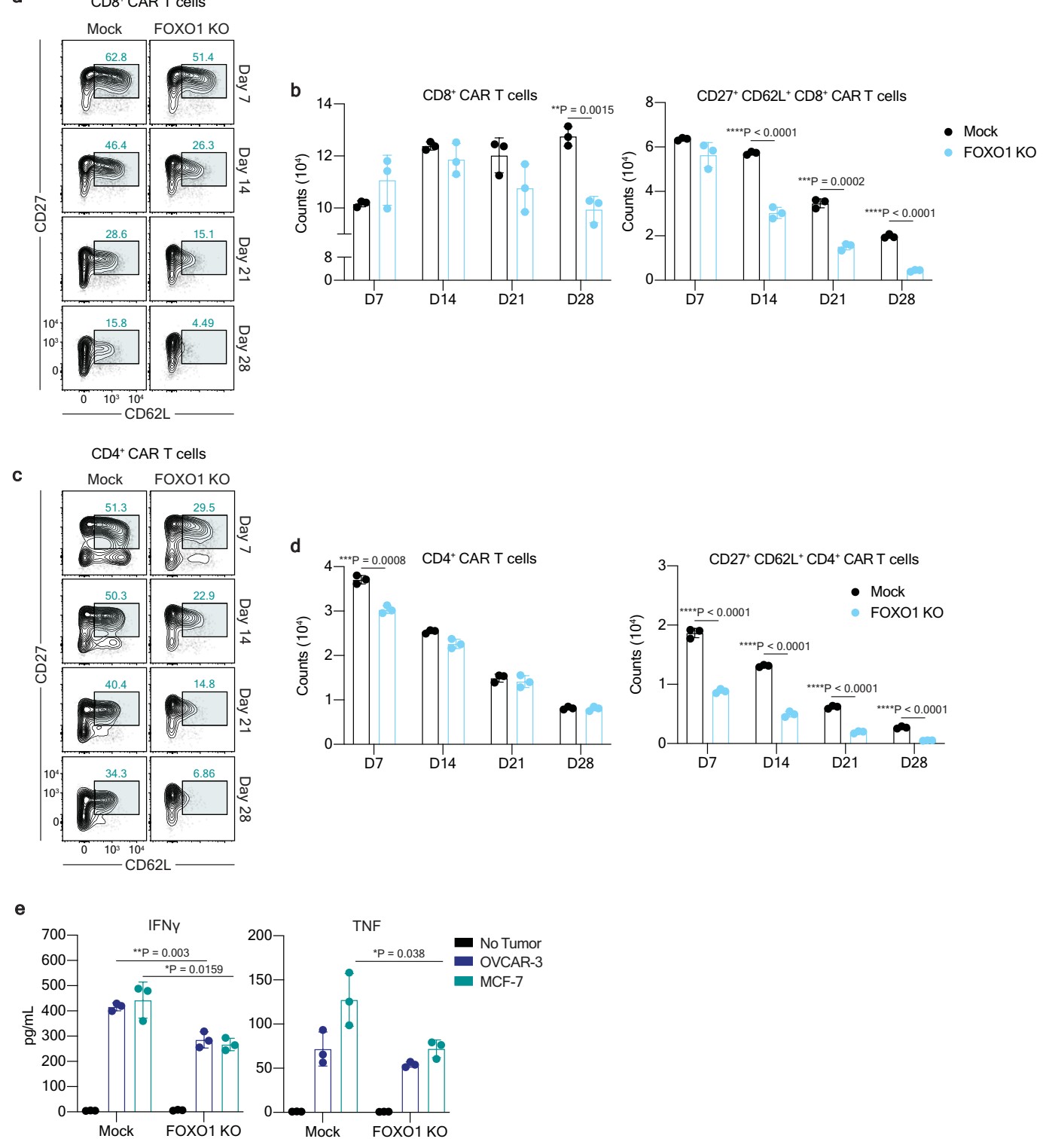

**Extended Data Fig. 9 | Phenotype and function of anti-Lewis Y CAR T cells following CRISPR/Cas9-mediated deletion of FOXO1.** Anti-Lewis Y CAR T cells deficient for FOXO1 were generated via CRISPR/Cas9 targeting. The expression of CD62L and CD27 on CD8+CAR+ T cells (**a-b**) and CD4+CAR+ T cells (**c-d**) were determined over 28 days in culture with IL-2. **a, c,** Representative flow cytometry plots of 3 independent donors. **b, d,** Numbers of total (left) or CD27+CD62L+ (right) CD8 + CAR T cells. Data represents the mean ± SD of technical triplicates from a representative experiment of n = 3 donors. Statistical significances determined by unpaired two-sided t test. **e,** CAR T cells were cocultured overnight at a 1:1 ratio with OVCAR-3 or MCF7 tumor cells for 16 h and the production of IFNγ or TNF determined by CBA. Data represents the mean ± SD of technical triplicates from a representative experiment of n = 3 donors. Statistical significance determined by unpaired two-sided t test, (*p < 0.05, **p < 0.01, ***p < 0.001, ****p < 0.0001. Source data are provided in the Source Data file.

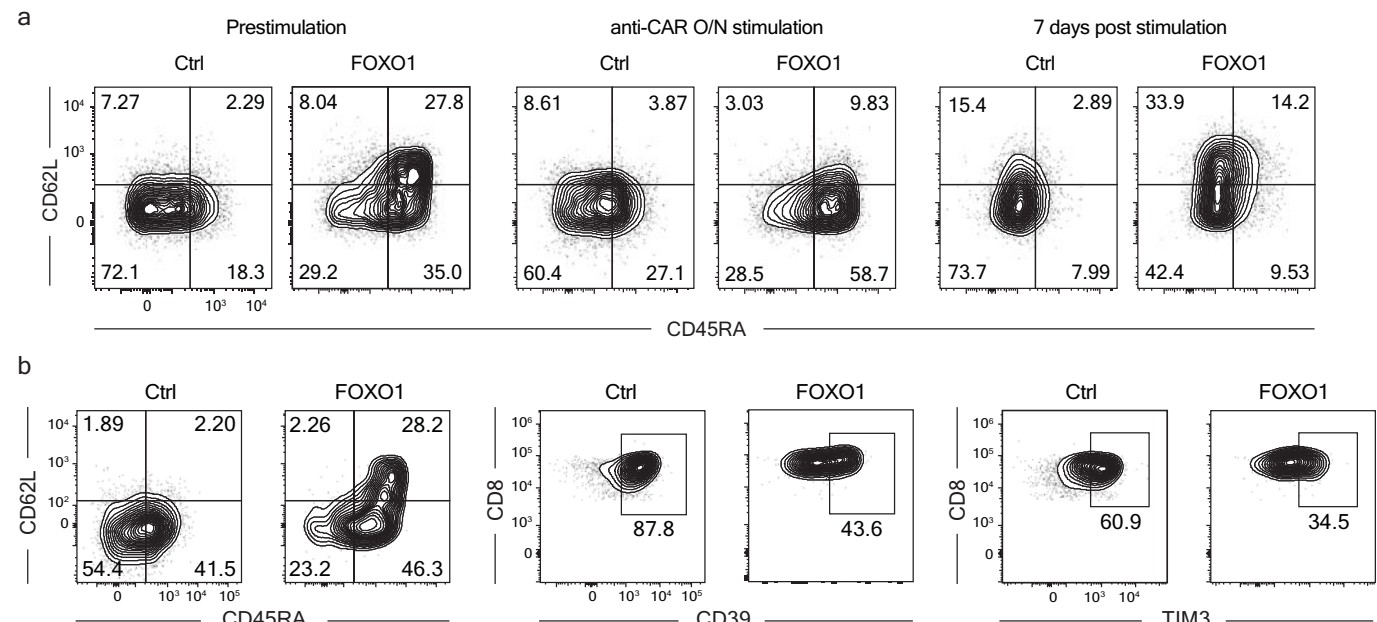

**Extended Data Fig. 10 | Phenotype of human CAR T cells expressing WT FOXO1 following repetitive stimulation cocultures. a**, Control and FOXO1 expressing CAR T cells were stimulated as per Extended Data Fig. 8n. Expression of CD45RA and CD62L on CD8⁺ CAR T cells is shown for a representative donor n = 3. **b**, Control and FOXO1 expressing CAR T cells were stimulated with MCF7 cancer cells for three weeks and the phenotype determined. Flow cytometry plots from a representative donor of n = 2.

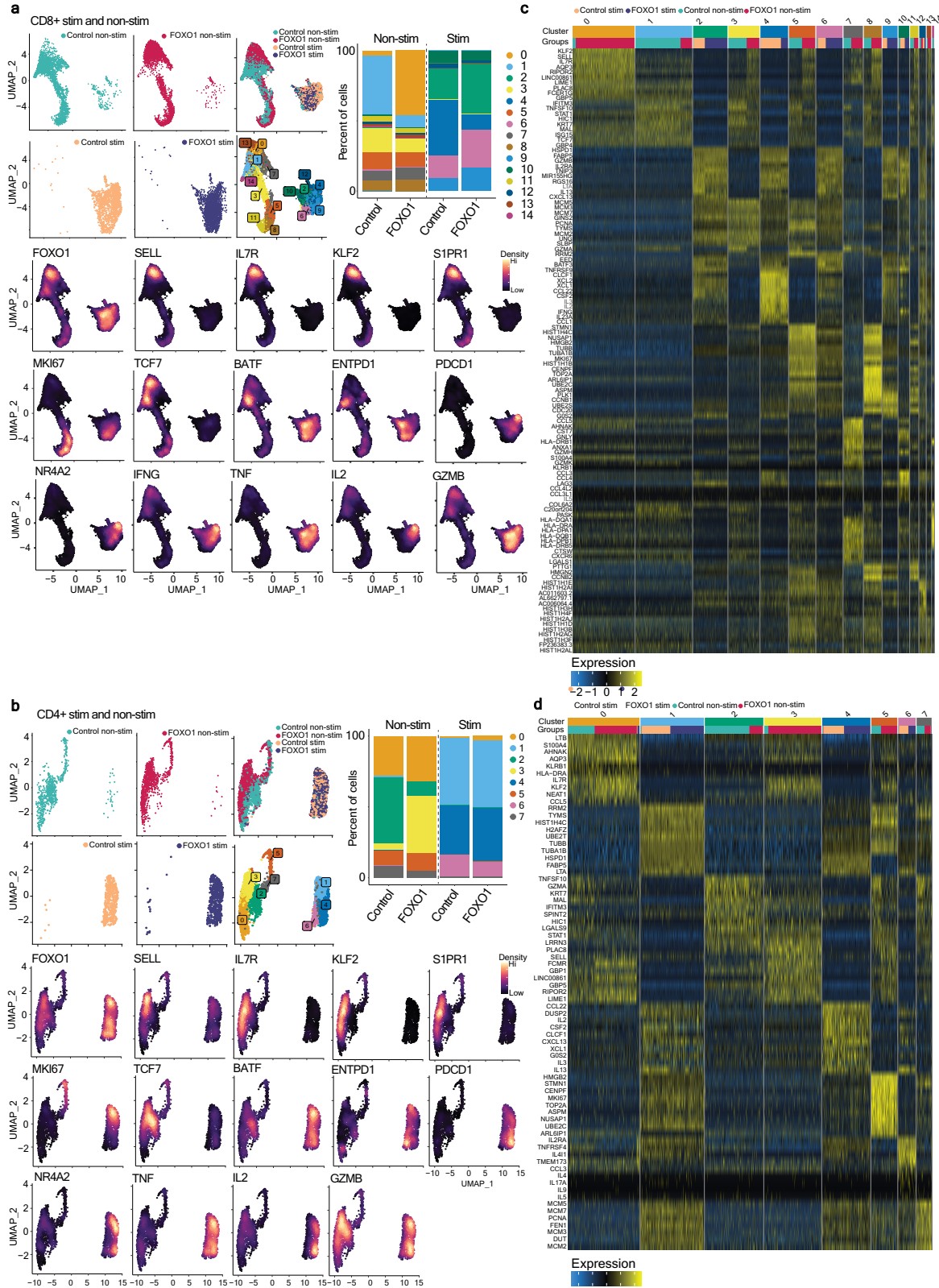

**Extended Data Fig. 11 | scRNA-seq analysis of control and FOXO1-expressing CAR T cells.** CAR T cells were either left non stimulated or cocultured with MCF7 tumor cells and analyzed by scRNA-Seq as per Fig. 3. **a-b,** UMAP plots, cluster composition and density plot of indicated genes shown for pooled stimulated and non-stimulated CD8+ (**a**) and CD4+ (**b**) CAR T cells. **c-d,** Heatmap indicating the expression of the top 10 differentially expressed genes in each (**c**) CD8+ and (**d**) CD4+ cluster.

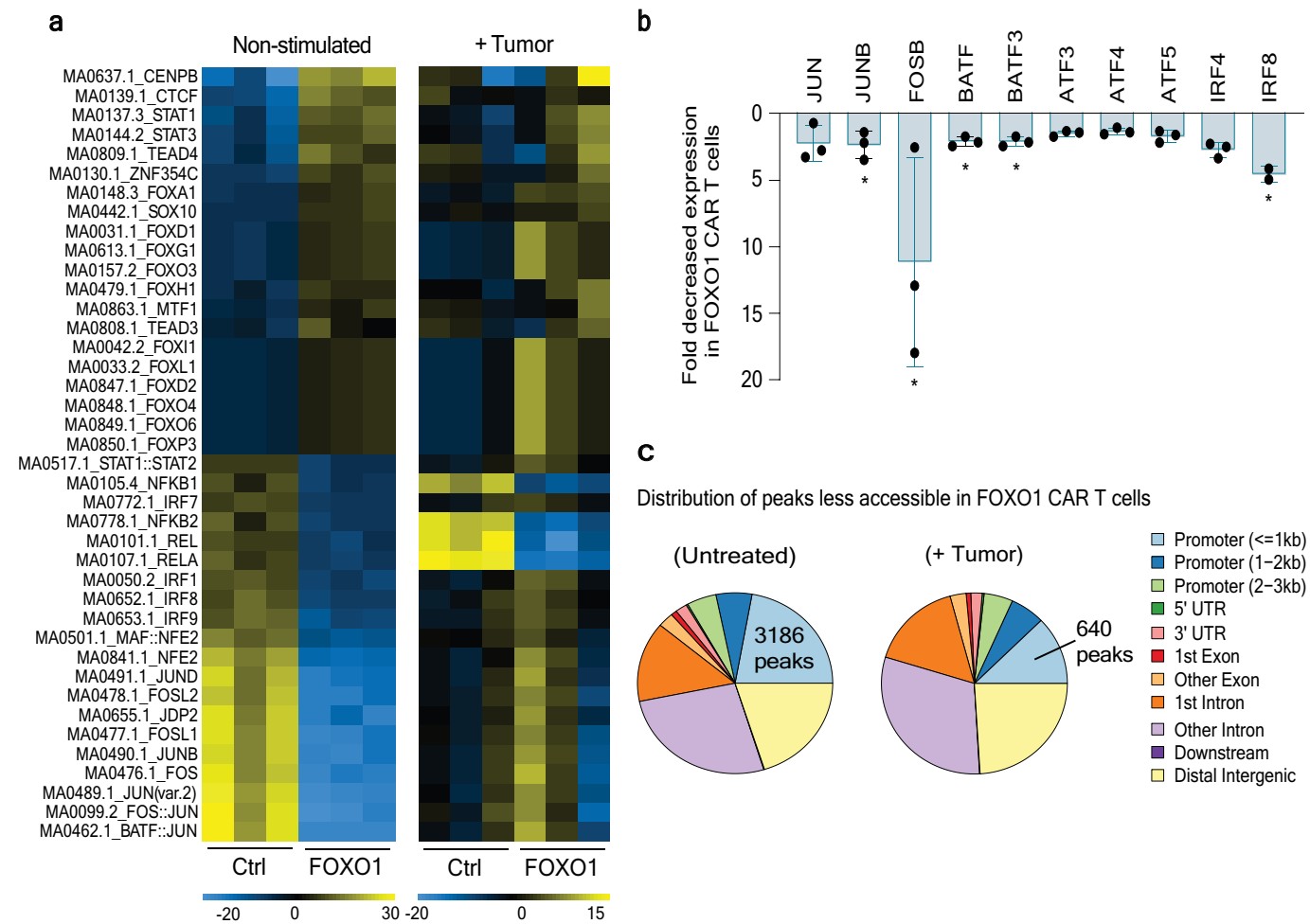

**Extended Data Fig. 12 | Epigenetic changes evoked in human CAR T cells by FOXO1 overexpression. a**, ChromVAR analysis of motifs (JASPAR) with increased or decreased accessibility in FOXO1-expressing CAR T cells. Heatmap depicts the top 20 motifs in each direction for non-stimulated cells and the same motifs after CAR activation. **b**, fold reduction in indicated transcription factor expression following FOXO1 overexpression in CD8⁺ CAR T cells analyzed as per Fig. 3e. **c**, Location of peaks with reduced accessibility in FOXO1-expressing T cells relative to controls prior to stimulation (left) and after MCF7 coculture (right). Source data are provided in the Source Data file.

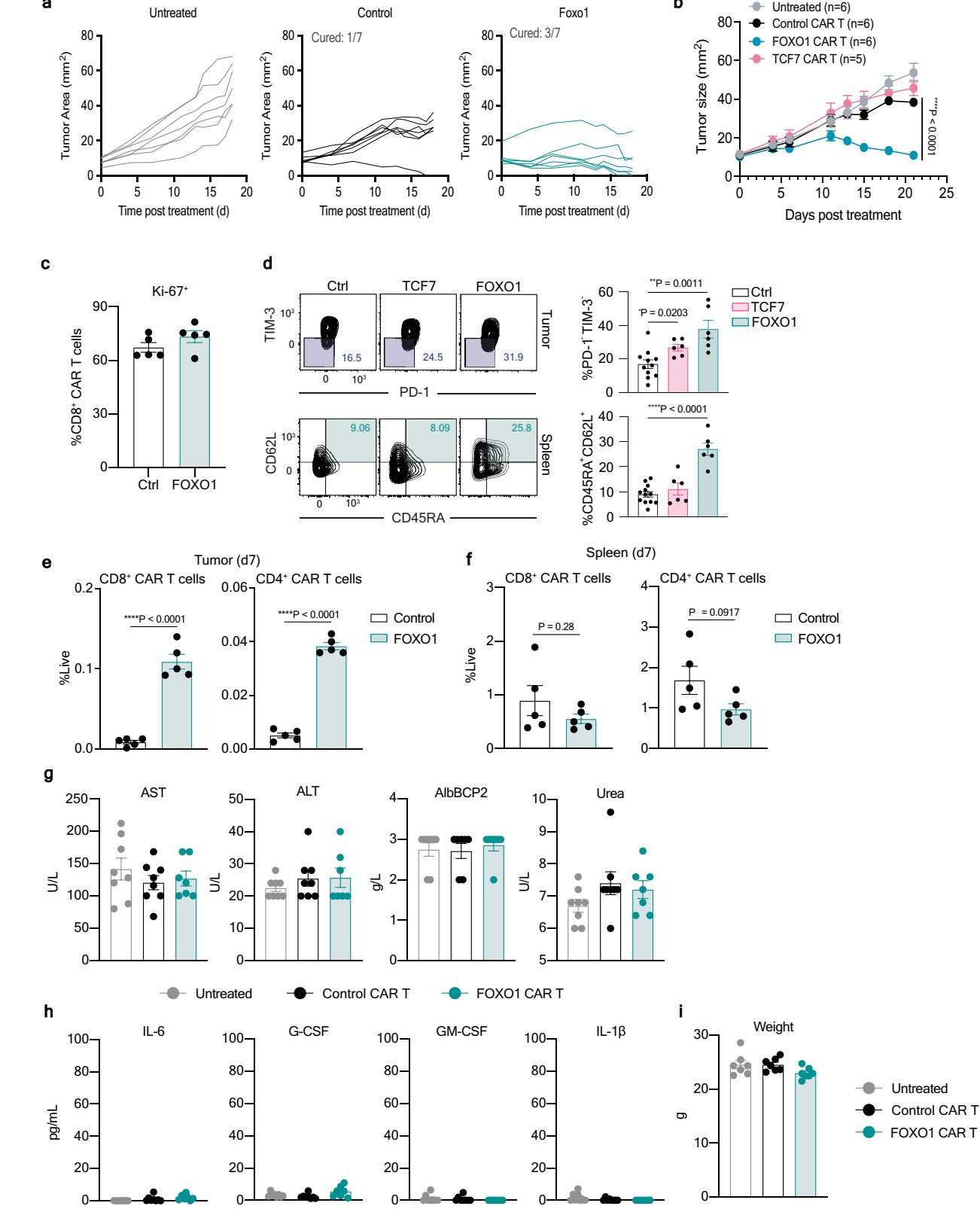

**Extended Data Fig. 13** | See next page for caption.

**Extended Data Fig. 13 | Phenotype and safety of FOXO1-expressing anti-Lewis Y CAR T cells in vivo. a**, Individual tumor growth curves of mice treated in Fig. 5d, 7 mice per group from a representative experiment of n = 2. **b.** Therapeutic efficacy of control, FOXO1 overexpressing or TCF7 overexpressing CAR T cells. Data presented as mean ± SEM of indicated numbers of mice representative experiment of n = 2. Statistic determined by Two-way ANOVA. **c**, Ki-67 expression of CD8$^+$ CAR T cells in the tumors of treated mice at day 12 post treatment. Data represents the mean ± SEM of 5 mice per group. **d**, Expression of PD-1 and TIM3 or CD45RA and CD62L in CD8$^+$ CAR T cells at day 13 post treatment. Representative FACS staining from concatenated samples (left) and data presented as mean ± SEM of n = 12 (Control) or n = 6 (Tcf7, Foxo1) (right).

Statistics determined by unpaired two-sided t test. **e**, **f**, Analysis of CAR T cell frequency in the tumors (**e**) and spleens (**f**) of treated mice at day 7 post treatment. Data represents the mean ± SEM of 5 mice per group. Statistics determined by unpaired two-sided t test. **g**, **h**, Expression of indicated enzymes (**g**) and cytokines (**h**) in serum at day 12 post treatment. Data represent the mean ± SEM of 8 (untreated, control) or 7 (FOXO1) independent mice. **i**, body weight of mice at day 19, experimental endpoint of Fig. 5d. Data represents the mean ± SEM of 7 mice per group from a representative experiment of n = 2. *p < 0.05, **p < 0.01, ****p < 0.0001. Source data are provided in the Source Data file.

| | |
|---|---|

# Reporting Summary

## Statistics

For all statistical analyses, confirm that the following items are present in the figure legend, table legend, main text, or Methods section.

| n/a | Confirmed | |
|---|---|---|
| ☐ | ☒ | The exact sample size ($n$) for each experimental group/condition, given as a discrete number and unit of measurement |
| ☐ | ☒ | A statement on whether measurements were taken from distinct samples or whether the same sample was measured repeatedly |
| ☐ | ☒ | The statistical test(s) used AND whether they are one- or two-sided *Only common tests should be described solely by name; describe more complex techniques in the Methods section.* |
| ☐ | ☒ | A description of all covariates tested |
| ☐ | ☒ | A description of any assumptions or corrections, such as tests of normality and adjustment for multiple comparisons |
| ☐ | ☒ | A full description of the statistical parameters including central tendency (e.g. means) or other basic estimates (e.g. regression coefficient) AND variation (e.g. standard deviation) or associated estimates of uncertainty (e.g. confidence intervals) |
| ☐ | ☒ | For null hypothesis testing, the test statistic (e.g. $F$, $t$, $r$) with confidence intervals, effect sizes, degrees of freedom and $P$ value noted *Give P values as exact values whenever suitable.* |
| ☒ | ☐ | For Bayesian analysis, information on the choice of priors and Markov chain Monte Carlo settings |
| ☒ | ☐ | For hierarchical and complex designs, identification of the appropriate level for tests and full reporting of outcomes |
| ☒ | ☐ | Estimates of effect sizes (e.g. Cohen's $d$, Pearson's $r$), indicating how they were calculated |

*Our web collection on statistics for biologists contains articles on many of the points above.*

## Software and code

Policy information about availability of computer code

| | |
|---|---|
| Data collection | Flow cytometry: BD FacsDiva version 8 (FlowJo LLC)<br>Seahorse assay: Seahorse XFe24 Bioanalyser (Agilent)<br>RNA-seq: CASAVA v1.8.2, Cutadapt v2.1, FastQC v0.11.6, RNA-SeQC v1.1.8<br>ATAC-Seq: Bcl2fastq (v2.20) |
| Data analysis | Flow cytometry: FlowJo version 10 (FlowJo LLC)<br>Differentially expressed gene analysis: EdgeR (https://bioconductor.org/packages/release/bioc/html/edgeR.html) v4.01<br>Gene set enrichment analysis: Enrichr (http://amp.pharm.mssm.edu/Enrichr)<br>General: Microsoft Excel 2010<br>Statistical analysis and data presentation: Graphpad Prism 9<br>Gene expression analysis: featureCounts, Rsubread 2.10.5; heatmaps: pheatmap R package v1.0.12, HISAT2 v 2.0.4<br>ATAC-Seq: MACS2 (v2.1.1), Genrich (v0.6.0), Homer v4.11, ChIPseeker (v1.8.6), IGV v2.7.0, findMotifsGenome, chromVAR |

For manuscripts utilizing custom algorithms or software that are central to the research but not yet described in published literature, software must be made available to editors and reviewers. We strongly encourage code deposition in a community repository (e.g. GitHub). See the Nature Portfolio guidelines for submitting code & software for further information.

## Data

Policy information about availability of data

All manuscripts must include a data availability statement. This statement should provide the following information, where applicable:
- Accession codes, unique identifiers, or web links for publicly available datasets
- A description of any restrictions on data availability
- For clinical datasets or third party data, please ensure that the statement adheres to our policy

Data availability statement has been provided in the manuscript as below. Additional raw data are available from the corresponding authors upon request.

Data availability Statement
The RNA-Sequencing and ATAC-Sequencing data that supports the findings of this study have been deposited in GEO NCBI under the accession code GSE225527 that contains the subseries GSE225521, GSE225522, GSE225523 and GSE225526. Reference gene sets were obtained from the MsigDB library for Hallmarks (https://www.gsea-msigdb.org/gsea/msigdb/human/collections.jsp), KEGG (https://www.genome.jp/kegg/kegg1.html), CHEA dataset 47, 48, 49, 50, or based upon previously published analyses of glycolysis signature 23, single-cell RNA sequencing derived T cell clusters in patients 11. Reference databases used are hg38 (https://www.ncbi.nlm.nih.gov/datasets/genome/GCF_000001405.26/), mm10 (Mus musculus genome assembly GRCm38 - NCBI - NLM (nih.gov)) and hg19 (https://www.ncbi.nlm.nih.gov/datasets/genome/GCF_000001405.13/). Source data are available within the paper (Microsoft® Excel® for Microsoft 365), supplementary information or available upon request from the authors.

## Human research participants

Policy information about studies involving human research participants and Sex and Gender in Research.

| | |
|---|---|
| Reporting on sex and gender | N/A |
| Population characteristics | Patients (aged >18 years) and had advanced solid tumours (local incurable or metastatic disease) and satisfied the eligibility criteria. |
| Recruitment | Patients were accrued onto the phase I clinical trial LeYPh1-02, trial number NCT03851146) following assessment of eligibility criteria as outlined on https://www.clinicaltrials.gov/ct2/show/study/NCT03851146. Eligibility criteria included advanced solid tumours which were (>10% LeY+ tumour cells positive) by IHC. |
| Ethics oversight | Ethics for the study was approved by the Peter MacCallum Cancer Centre Human Research Ethics committee. |

Note that full information on the approval of the study protocol must also be provided in the manuscript.

# Field-specific reporting

Please select the one below that is the best fit for your research. If you are not sure, read the appropriate sections before making your selection.

☒ Life sciences   ☐ Behavioural & social sciences   ☐ Ecological, evolutionary & environmental sciences

For a reference copy of the document with all sections, see nature.com/documents/nr-reporting-summary-flat.pdf

# Life sciences study design

All studies must disclose on these points even when the disclosure is negative.

| | |
|---|---|
| Sample size | Experiments were performed with sufficient power to achieve statistical significance based upon an effect size of 30%, which would have been deemed clinically significant. All therapeutic experiments were performed with a minimum of 3 mice per group. |
| Data exclusions | No data were excluded from the manuscript. |
| Replication | All experiments were replicated in at least 2 independent experiments. |
| Randomization | Mice were randomized prior to treatment according to tumor size to ensure all groups had equivalent tumor burden prior to therapy. Groups were age and sex matched. Randomization did not apply to in vitro studies because experiments were conducted with a common source of biological material e.g. the same PBMCs. |
| Blinding | Data was not blinded. The same investigators performed and analyzed experiments and so blinding was not possible. |

# Reporting for specific materials, systems and methods

We require information from authors about some types of materials, experimental systems and methods used in many studies. Here, indicate whether each material, system or method listed is relevant to your study. If you are not sure if a list item applies to your research, read the appropriate section before selecting a response.

## Materials & experimental systems

| n/a | Involved in the study |
|-----|----------------------|
| ☐ | ☒ Antibodies |
| ☐ | ☒ Eukaryotic cell lines |
| ☒ | ☐ Palaeontology and archaeology |
| ☐ | ☒ Animals and other organisms |
| ☒ | ☐ Clinical data |
| ☒ | ☐ Dual use research of concern |

## Methods

| n/a | Involved in the study |
|-----|----------------------|
| ☒ | ☐ ChIP-seq |
| ☐ | ☒ Flow cytometry |
| ☒ | ☐ MRI-based neuroimaging |

## Antibodies

| | |
|---|---|
| Antibodies used | All antibodies used in the study were obtained from commercial suppliers (BD Pharmigen, Cell Signaling, eBioscience, Invitrogen, Thermo Scientific or Biolegend). A list of relevant information on antibodies is provided in the Supplementary Table |
| Validation | All antibodies were validated by the supplier. Relevant information can be found in the Supplementary Table |

## Eukaryotic cell lines

Policy information about cell lines and Sex and Gender in Research

| | |
|---|---|
| Cell line source(s) | The murine colon adenocarcinoma MC38-Her2 cell line was generated from cells obtained from Dr. Jeff Schlom (National Institute of Health, Maryland, USA). The mouse breast carcinoma E0771-Her2 cell line were generated from cells obtained from Prof. Robin Anderson (Olivia Newton-John Cancer Centre, Victoria, Australia). These cell lines were not obtained from a commercial source. PA317, GP+e86 and HEK293T, OVCAR-3 and MCF7 tumor cell lines were obtained from the American Type Culture Collection. |
| Authentication | Cell lines were not authenticated but were uitiized within 10 passages of a master stock |
| Mycoplasma contamination | All lines were tested negative for mycoplasma contamination |
| Commonly misidentified lines (See ICLAC register) | None of the cell lines are listed on the ICLAC database |

## Animals and other research organisms

Policy information about studies involving animals; ARRIVE guidelines recommended for reporting animal research, and Sex and Gender in Research

| | |
|---|---|
| Laboratory animals | C57BL/6 mice, NSG and C57BL/6 human-Her2 (hHer2) transgenic mice were utilized where indicated. Mice were used between 6-16 weeks of age.<br><br>Housing was as follows;<br>Cage type – Allentown IVC<br>Bedding – irradiated corncob<br>Diet – Barastoc irradiated commercial rat and mouse pellets<br>Light cycle – 10 hour dark/14 hour light with a half hour sunrise and a half hour sunset period<br>Ambient temperature is set at 20oC +/- 1degree<br>Relative humidity is 40% |
| Wild animals | This study did not involve wild animals. |
| Reporting on sex | Studies utilizing E0771-Her2 and OVCAR-3 cells were performed in female mice. Studies using MC38-Her2 were performed in male mice. |
| Field-collected samples | No field collected samples were used in the study. |
| Ethics oversight | Ethics oversight was performed by the Peter MacCallum Cancer Centre Animal Experimentation Ethics Committee (AEEC) at the Peter MacCallum Cancer Centre. |

Note that full information on the approval of the study protocol must also be provided in the manuscript.

# Flow Cytometry

## Plots

Confirm that:

☒ The axis labels state the marker and fluorochrome used (e.g. CD4-FITC).

☒ The axis scales are clearly visible. Include numbers along axes only for bottom left plot of group (a 'group' is an analysis of identical markers).

☒ All plots are contour plots with outliers or pseudocolor plots.

☒ A numerical value for number of cells or percentage (with statistics) is provided.

## Methodology

| | |
|---|---|
| Sample preparation | Blood was collected via submandibular or retroorbital bleed into tubes containing EDTA prior to euthanasia. Blood and spleen samples were treated twice or once respectively with ACK lysis buffer before staining for flow cytometry. Tumors were digested in SAFC DMEM media with 0.01 mg/mL DNase (Sigma Aldrich) and 1 mg/mL type IV collagenase for 30 minutes at 37oC. Following digestion, tumor samples were filtered twice through a 70 μm filter to create a single cell suspension and resuspended in Fc block prior to staining for analysis by flow cytometry. |
| Instrument | FACS data were obtained on a BD FACS Symphony A5 and BD LSRFortessa X-20 from the Peter MacCallum Cancer Centre flow cytometry core facility. |
| Software | Data was analyzed using Flowjo software 10.8.1 |
| Cell population abundance | For experiments where cells with FACS sorted, re-analysis was performed to confirm sort purity |
| Gating strategy | FSC/SSC gate was first used to gate on the morphology of leukocytes. A singlet gates was next used (FSC-A vs FSC-h) to exclude doublets followed by a viability gate (Fixable Yellow or DAPI) to excluded dead cells. CAR T cells were identified using a T cell lineage marker (Thy1.2 or CD3) and transduction marker (mCherry, NGFR or FLAG). |

☒ Tick this box to confirm that a figure exemplifying the gating strategy is provided in the Supplementary Information.

