## [Peer Review File · Nature]

Manuscript Title: FOXO1 enhances CAR T cell stemness, metabolic fitness and efficacy

Reviewer Comments & Author Rebuttals

Reviewer Reports on the Initial Version:

Referees' comments:

Referee #1 (Remarks to the Author):

In this paper, Chan et al. present a series of experiments detailing how the manipulation of FOXO1 can be used to enhance “stemness” of CAR-T cells, augment the number and proportion of polyfunctional CAR-Ts, decrease the expression of checkpoint molecules such as PD-1, TIM3 and LAG3, and lead to enhanced elimination of tumors in mouse models. While many of the findings are potentially interesting, they seem incompletely explored leading to a number of major concerns:

1. There are no conclusive data in the first sets of figures that show that expression of constitutively active FOXO1 is superior to other transcription factors, particularly Tcf7. Why then does the second half of the paper focus on FOXO1? Are we to understand that targeting FOXO1 is “better”? The message is very unclear.
2. Overexpression of the checkpoint molecules cited above is consistent with exhaustion, but as the authors undoubtedly know, these molecules are induced by activation. Their expression is not an indicator of exhaustion per se. More detailed studies would be needed to demonstrate that exhaustion is prevented.
3. Surprisingly, there is no acknowledgement in the results that the human CAR T cell experiments were done in immunodeficient mice. What concerns/caveats does that raise. How might the absence of an immune system and the fact that the recipient mice are irradiated confound the results?
4. I would have liked to see more mechanistic exploration of why constitutive FOXO1 activation in human T cells leads to a different result than in mice, but that overexpression of FOXO1 in human T cells mimics constitutive expression in mice. What is the level of overexpressed FOXO1 compared to endogenous at rest and how much is in the nucleus after activation?

Other specific points:

Lines 77-80 refer to solid tumors but one of the cited references, number 15, is about leukemia.

Please reference the statement in line 358 about “. . . the critical importance of FOXO1 in maintaining CAR T cell stemness . . . “

In presenting figure 1a, the authors never mention the use of IL-7 in addition to IL-15 or IL-2.

Supplementary figure 1a-b. This is pretty low-level overexpression. Please comment and please show the controls for each transcription fact.

In the model in figure 2g-i don't the CAR-T cells attack the endogenous HER2 expressing tissues?

Referring to figure 6 if n=7 in panel a, why aren't there 7 data points in b-e? And why are there 8 data points in some part of panel f, but 7 in other parts?

Referee #2 (Remarks to the Author):

CAR-T and Adoptive Cellular Therapy (ACT) continue to emerge as a potentially powerful means to treat cancer. This has been spurred on by initial successes and FDA approvals. However, it is clear that there is still much work to be done in terms of developing this modality. Specifically, CAR-T cells have yet to demonstrate significant efficacy against solid tumors. Likewise, it is clear that efficacy is related to persistence of "memory" like cells which can provide ongoing effector cell generation. In this report the author's provide evidence that engineering cells to over-express FOXO1 can lead to more robust CAR-T cells fit to overcome these hurdles.

Overall, this is a very exciting and important work that will provide great insight to the field. The paper itself is exceedingly clear and well written and the experiments and data are thoughtful and robust. I enjoyed reading this paper and learned a lot. Specifically, the work is very comprehensive in relating gene expression with metabolism with function. There are just a couple of issues that it would be nice for the authors to address:

First, there is a lot of functional data comparing the Foxo1-ADA with Tcf7. However, it would be nice to see some of the mechanistic comparisons between the two. For example, in Figures 3f and 4J & K how does Tcf7 compare? Also, is there any data regarding the Tcf7 overexpression in the human cells?

Second, the differences between the overexpression of the Foxo1-ADA and Foxo1 in the human cells is important. The authors suggest that the differences may be due to higher levels of overexpression as a consequence of using a different promoter. This hypothesis should be tested and the authors should consider performing more experiments to hone in to the cause of the differential effects of the Foxo1-ADA versus the Foxo1 overexpression in the human cells.

Referee #3 (Remarks to the Author):

Summary of the key results: the authors propose FOXO1 constitutive activation/overexpression as a potential strategy to address limitations of CAR-T therapy for solid tumours.

Originality and significance: The importance of FOXO1 to CAR-T functionality has been elegantly described by Klebanoff et al in JCI Insight in 2015 in the context of AKT inhibition, with similar conclusions to this paper about the role of FOXO1 in promoting superior phenotype and functionality in a CD19CAR-T model. JCI Insight. 2017 Dec 7;2(23):e95103.doi: 10.1172/jci.insight.95103.

Data & methodology: valid, good quality, well presented data

Appropriate use of statistics and treatment of uncertainties:yes

References: in my opinion need to add Klebanoff paper and talk about AKT inhibition wrt FOXO1 as an alternative (non genotoxic) way to effect the same cell 'fitness' as an engineered approach, and to show how it differs/why your approach is better.

Specific comments:

1. Figure 2h: shows difference in tumour growth at day 10-12, but in supplementary figure 2 it looks like majority tumours are only transiently controlled and animals all dead by ~Day 28-35. Is this the best we can expect- why do no animals fully reject tumours? Initial CAR expansion controls and then the CARs stop dividing? Or the TME switches them off, irrespective of FOXO?
2. Figure 3g/h and line 660: the inference that FOXO1-ADA makes CAR-T cells LN resident, is this just a feature of better in vivo expansion with FOXO1-ADA rather than being a reprogramming/trafficking event? What other features of lymphoid resident cells could you test for to help support this statement?
3. Line 583: should this read 'does NOT preclude epigenetic sic..'
4. Line 606-608: cannot make any inference about CAR-T toxicity from a model with no functioning immune system (NSG) so this should be significantly softened/removed.
5. Line 693: this is fundamental. Is the difference between murine and human FOXO-ADA due to higher transcripts and expression in human system which disables CAR-T into being unable to perform effector functions? Can you show the data as to whether there were higher levels of transcripts? If this is simply a question of FOXO1-ADA 'dose' then hypothesis should be quite easy to test- to titrate expression to find 'sweet spot' where preserved stemness and function co-exist, as they did in the murine model.
6. Line 693 and Figure 5: When you use FOXO1 overexpression instead of FOX1-ADA in the human model, you show that many of the perceived benefits of FOXO1 overexpression are reversed on stimulation. The critical point to address is whether the cells then revert back to desirable

phenotypes when stimulation state is removed i.e. does the continued overexpression of FOXO1 confer a 'plastic' state where cells shuttle between memory and effector depending on the presence of activation? If not, and only transient benefit from FOXO1 gene engineering, could this same benefit be more simply effected by using an AKT inhibitor in vitro as per Klebanoff?

7. Figure 6 legend: cannot say that FOXO1 improves 'persistence' if testing in an NSG model and only have data out to day 12 or so? Maybe you mean expansion?

8. Line 789: how are you defining 'exhaustion' here? I can't see all the facets of T-cell exhaustion shown/discussed here other than preserved functionality.

9. Figure 6d: lower Tim3/PD-1 in FOXO1 condition - is this the same in the tumour and spleen?

10. Figure 6i: patient cells - Is the expression of FOXO1 in cis with CAR? Linked by 2A/IRES? Is there any functional data here to mirror what you have shown in healthy cells?

Conclusions: as per specific comments above

Suggested improvements: as per specific comments above

Clarity and context: This is clearly written, but needs more context as per comments above wrt AKTi, and there are several important points to address above to fully realise the potential of this nice piece of work. I would be happy to review a revised version.

Author Rebuttals to Initial Comments:

Referees' comments:

Referee #1 (Remarks to the Author):

In this paper, Chan et al. present a series of experiments detailing how the manipulation of FOXO1 can be used to enhance “stemness” of CAR-T cells, augment the number and proportion of polyfunctional CAR-Ts, decrease the expression of checkpoint molecules such as PD-1, TIM3 and LAG3, and lead to enhanced elimination of tumors in mouse models. While many of the findings are potentially interesting, they seem incompletely explored leading to a number of major concerns:

1. There are no conclusive data in the first sets of figures that show that expression of constitutively active FOXO1 is superior to other transcription factors, particularly Tcf7. Why then does the second half of the paper focus on FOXO1? Are we to understand that targeting FOXO1 is “better”? The message is very unclear.

Response: We understand the concern of the reviewer. In the original submission we had performed a series of experiments comparing FOXO1, TCF7 and ID3 in murine CAR T cells but not in the context of human CAR T cells. Therefore, to address this question, we have performed a series of new experiments using human CAR T cells transduced with either TCF7 or ID3 and compared the phenotype and function to CAR T cells overexpressing FOXO1. First, we investigated the impact of TCF7 and ID3 on the phenotype of CAR T cells *in vitro* and compared to FOXO1 overexpressing CAR T cells. These experiments show that FOXO1 overexpression results in an enrichment of the CD45RA⁺CD62L⁺ population and this effect is significantly more profound than observed with either TCF7 or ID3. This new data is presented in new **Figure 4B** and referred to in the following text located on line 579.

“We next compared the impact of WT FOXO1, TCF7 and ID3 on the phenotype of anti-Lewis Y CAR T cells. FOXO1 was significantly more effective in the induction of the CD45RA⁺CD62L⁺ population of CD8⁺ CAR T cells relative to TCF7 and ID3 (Fig. 4b) and indeed the increased proportion of CD45RA⁺CD62L⁺ cells following FOXO1 expression was reproducible across multiple donors (Fig 4c).”

Moreover, whilst we observed that FOXO1 overexpression was able to significantly reduce the expression of TIM3 and CD39 on CD8⁺ CAR T cells both during CAR T cell *in vitro* expansion (**Figure 4B**) and after coculture with MCF7 tumor cells (**Supplementary Figure 6F**), neither TCF7 nor ID3 were able to mediate this effect. This data is referred to in the following text located on line 592.

Notably, the reduced expression of CD39 mediated by FOXO1 was not recapitulated by the overexpression of either TCF7 or ID3 either in T cell only cultures (Fig. 4b) or following tumor cell coculture (Supplementary Fig. 6f), providing further evidence for an enhanced capacity of FOXO1 to favorably modulate CAR T cell phenotype relative to these factors.”

Given the observation that ID3 overexpression resulted in a decreased proportion of CD45RA⁺CD62L⁺ cells and an increased proportion of CD39⁺TIM3⁺ cells, we subsequently focused on comparisons between FOXO1- and TCF7-overexpressing CAR T cells. In our original submission our data indicated that FOXO1 overexpression resulted in a positive enrichment for genes associated with less differentiated cells identified to correlate with improved responses to immune checkpoint blockade in patients (Sade-Feldmann et al. 2018, Cell; PMID 30388456). In our revised manuscript we show that TCF7 overexpression did not recapitulate this effect and this new data is shown in new **Supplementary Figure 6G**. We also used the RNA-sequencing data to show that FOXO1 overexpression decreased the expression of immune checkpoints (**Supplementary Figure 6I**) and genes associated with exhaustion (**Supplementary Figure 6L**) to a significantly greater extent than with TCF7 overexpression. These data are referred to in the following passages of text on lines 618 and 647.

“As expected, this analysis revealed that FOXO1 overexpressing CD8⁺ CAR T cells exhibited increased expression of genes associated with less differentiated T cells relative to their control counterparts (Fig. 4e). This effect was significantly more pronounced than observed following TCF7 overexpression, confirming that FOXO1 instigated a broad transcriptional program that enhanced the stem-like state of CAR T cells (Supplementary Fig. 6g). Moreover, WT FOXO1 overexpressing CD8⁺ CAR T cells exhibited decreased expression of immune checkpoints relative to control counterparts and this effect was more marked than with TCF7 expressing CAR T cells (Supplementary Fig. 6h-i)”

“Lastly, we utilized RNA-sequencing to address the question of whether FOXO1 overexpression could protect CAR T cells from exhaustion. In the steady state, we observed that FOXO1 overexpression resulted in a significant negative enrichment of genes associated with exhaustion (67), an effect that was not observed following TCF7 overexpression (Supplementary Fig 6l).”

We also compared the capacity of TCF7 overexpression to modulate the metabolic profile of CAR T cells using a Seahorse assay. Whilst FOXO1 overexpression significantly enhanced basal respiratory capacity, maximal respiratory rate, and spare respiratory capacity, TCF7 overexpression made minimal impact on these parameters. This new data is presented in new **Figure 4H** and referred to in the following text on line 640.

“Notably, there were minimal changes to these parameters in the context of TCF7 overexpressing CAR T cells, highlighting that FOXO1 overexpression was a superior strategy to modulate CAR T cell metabolism (Fig. 4h).”

Lastly, we compared the impact of TCF7, ID3 or FOXO1 overexpression on the number and phenotype of CAR T cells *in vivo*. FOXO1 was the only transcription factor that significantly enhanced the efficacy of CAR T cells and was associated with a significant increase in the number of CAR T cells in the spleen and blood of treated mice. Moreover, whilst FOXO1 overexpression led to a significant enhancement of the proportion of CD45RA⁺CD62L⁺ cells, this was not observed following TCF7 overexpression. This new data is shown in new **Figure 6a-c** and **Supplementary Figure 10b** and is referred to in the following text located on lines 732 and 741.

“To compare the impact of TCF7, ID3 and FOXO1 overexpression in vivo we treated OVCAR-3 tumor bearing mice with CAR T cells expressing each transcription factor. Strikingly, FOXO1 was the only transcription factor that enhanced CAR T cell efficacy as indicated by a significant decrease in tumor size (Fig. 6a), which was associated with a significant increase in the number of CD8⁺ and CD4⁺ CAR T cells in the blood and spleen of treated mice (Fig. 6b-c).”

“In both tumors and spleens WT FOXO1 overexpressing CD8⁺ CAR T cells exhibited a less differentiated phenotype (Fig. 6f-g). Tumor-infiltrating FOXO1 overexpressing CD8⁺ CAR T cells also exhibited reduced expression of the exhaustion markers PD-1 and TIM3 relative to control CAR T cells but exhibited a similar proliferative capacity as indicated by Ki-67 expression (Fig. 6h, Supplementary Fig. 10b). Notably, these phenotypic differences were not observed following TCF7 overexpression, confirming the enhanced capacity of FOXO1 to favorably modulate CAR T cell phenotype and function (Supplementary Fig. 10c-d).”

2. Overexpression of the checkpoint molecules cited above is consistent with exhaustion, but as the authors undoubtedly know, these molecules are induced by activation. Their expression is not an indicator of exhaustion per se. More detailed studies would be needed to demonstrate that exhaustion is prevented.

Response: To further interrogate the impact of FOXO1 expression we have performed additional experiments showing that FOXO1-overexpressing CAR T cells maintain a less differentiated phenotype (CD45RA⁺CD62L⁺ and CD45RA⁻CD62L⁺) relative to control CAR T cells upon chronic stimulation with Lewis Y antigen positive tumor cells. These experiments reveal that FOXO1-expressing CAR T

cells also maintain lower expression of CD39 and TIM-3 under these conditions. This new data is shown in **Supplementary Figure 8b** and referred to in the following text on line 609.

“Similarly, when control or FOXO1-expressing CAR T cells were exposed to tumor cells over a period of 3 weeks FOXO1 CAR T cells maintained an enriched proportion of CD45RA⁺CD62L⁺ CAR T cells as well as reduced expression of TIM3 and CD39 (Supp Figure 8b).”

To complement these functional assays, we further interrogated the RNA-Seq dataset and observed that FOXO1 overexpressing CAR T cells exhibited a significant negative enrichment for genes associated with exhaustion. Furthermore, we observed that this negative enrichment was maintained in CAR T cells that had been activated and then rested. This new data is referred to in new **Supplementary Figure 6l-m** and referred to in the text on line 647.

“Lastly, we utilized RNA-sequencing to address the question of whether FOXO1 overexpression could protect CAR T cells from exhaustion. In the steady state, we observed that FOXO1 overexpression resulted in a significant negative enrichment of genes associated with exhaustion (67) relative to control CAR T cells, an effect that was not observed following TCF7 overexpression (Supplementary Fig 6l). We next evaluated this in the context of stimulation and resting. We again observed that prior to stimulation FOXO1-expressing CAR T cells exhibited reduced expression of exhaustion-related genes (Supplementary Fig. 6m). Following stimulation through the CAR both control and FOXO1-expressing CAR T cells upregulated these genes such that there was no significant difference between the groups. However, after a period of rest, the negative enrichment for this gene set was restored in FOXO1 expressing CAR T cells, highlighting that these cells are protected from the transcription of genes associated with exhaustion.

We also further analyzed the RNA-Seq data to specifically interrogate for transcription factors associated with exhaustion. To do this we compared the epigenetic signatures observed in FOXO1-expressing CAR T cells to those reported in a tonic signaling CAR exhaustion model (Lynn et al. 2019, *Nature*). This analysis revealed that FOXO1 overexpression resulted in a significant negative enrichment for factors observed to be significantly increased in exhausted CAR T cells. This new data is shown in new **Figure 5D** and is referred to in the following text on line 706.

“Indeed, FOXO1 overexpression resulted in reduced expression of transcription factors previously observed to be upregulated in exhausted CAR T cells (70), including members of the AP-1 family such as IRF8 and FOSB (Fig. 5d).

Lastly, to assess the question of how FOXO1 overexpression significantly reduces the gene signatures associated with exhaustion we performed a scRNA-seq analysis of control and FOXO1 expressing CAR T cells before and after activation with tumor cells. This revealed that FOXO1 overexpression resulted in the expansion of a cluster of cells characterized by high expression of KLF2, SELL and IL7R. Notably this subset of cells exhibited reduced expression of genes associated with exhaustion relative to a relevant cluster of control CAR T cells. This new data is presented in new **Figures 4I and 4K** and is referred to in the following text on line 666.

This confirmed that FOXO1 expression enhanced the proportion of CD8⁺ CAR T cells with a more ‘stem-like’ appearance, notably through the enrichment of a cluster (designated cluster 1) that was characterized by high expression of KLF2, SELL and IL7R (Fig. 4i, Supplementary Fig. 9c). This enhanced proportion of cluster 1 cells was offset by a reduction in the proportion of a cluster of cells (designated cluster 0) characterized by high expression of IFITM3 and TNFSF10 (Supplementary Fig. 9c). Indeed, comparing FOXO1 expressing CAR T cells within cluster 1 to control CAR T cells within cluster 0 revealed a significant enrichment for genes associated with less differentiated cells and reduced expression of genes associated with glycolysis and exhaustion, corroborating our results from bulk RNA-sequencing (Fig. 4k).

3. Surprisingly, there is no acknowledgement in the results that the human CAR T cell experiments were done in immunodeficient mice. What concerns/caveats does that raise. How might the absence of an immune system and the fact that the recipient mice are irradiated confound the results?

Response: We agree with the reviewer's concerns with regards to the use of immunodeficient mice to assess safety although this is the only model available for the testing of human CAR T cells. Therefore, to complement this we have performed new experiments in the syngeneic Her2 model that recapitulates the existing safety data from NSG mice. That is to say that with FOXO1-ADA expressing murine CAR T cells we also observed no significant changes to the levels of enzymes associated with liver and kidney function, cytokines associated with cytokine release syndrome or overt signs of pathology as determined by histological staining of brain, liver and lungs. This new data is shown in **New Supplementary Figure 2F-I** and referred to in the following text on line 532.

“Importantly, the increased therapeutic efficacy observed with Foxo1-ADA overexpressing CAR T cells was not associated with overt signs of toxicity (Supplementary Figure 2f-i), highlighting the clinical potential of this approach.”

4. I would have liked to see more mechanistic exploration of why constitutive FOXO1 activation in human T cells leads to a different result than in mice, but that overexpression of FOXO1 in human T cells mimics constitutive expression in mice. What is the level of overexpressed FOXO1 compared to endogenous at rest and how much is in the nucleus after activation?

Response: To address this question regarding the differences between human and mouse systems, we first assessed the expression of FOXO1 in each model via qRT-PCR. This new data is shown in **New Supplementary Figure 1B-C** and indicates that the overexpression of FOXO1 was significantly more pronounced in the human CAR T cell experiments where transgene expression was driven by the EF1 α promoter compared to the murine system where transgene expression was driven by the retroviral LTRs.

We therefore hypothesized that the differences may simply be explained by the increased expression of wild-type FOXO1 achieved in the human system. To test this hypothesis, we generated new lentiviral constructs whereby either FOXO1-ADA or wild-type FOXO1 was driven by the PGK or CMV promoters. According to the literature, EF1 α was predicted to be the strongest promoter followed by PGK and then CMV being the weakest promoter (PMID 32706785).

In our hands, transgene expression was markedly lower with the CMV promoter enabling us to investigate the FOXO1 gene dosage effect on the phenotype of CAR T cells. These experiments revealed that whilst FOXO1-ADA expression significantly enhanced the proportion of CD45RA⁺CD62L⁺ cells when driven by any promoter, the impact of WT FOXO1 on the proportion of CD45RA⁺CD62L⁺ cells was diminished as the promoter strength decreased. Therefore, we believe that the lack of phenotype observed with WT FOXO1 in the murine system was due to the low level of transgene overexpression achieved in the murine system, however other species differences also cannot be discounted. We next assessed whether FOXO1-ADA expressing human CAR T cells would produce cytokines when expression was driven by the weaker CMV promoter (given our previous data indicating that FOXO1-ADA expression driven by the EF1 α promoter almost completely attenuated cytokine production). These experiments revealed that FOXO1-ADA significantly reduced production of IFN γ and TNF regardless of the promoter strength. These new data support our original data that engineering CAR T cells to express WT FOXO1 from a strong promoter leads to the generation of CAR T cells with the most favorable characteristics, i.e. enhanced proportions of naïve like cells and the capacity to produce cytokines upon activation.

This new data is shown in **New Supplementary Figure 5F-I** and referred to in the following sections of text on line 558 in the results section.

Given the phenotypic differences observed between the murine and human systems following overexpression of the constitutively active variant of FOXO1, we investigated whether this was due to differences in expression levels of the transgene between the two models. Indeed, overexpression of FOXO1 in human CAR T cells was more pronounced relative to murine counterparts with approximately a 5- and 2-fold increase in human FOXO1/ murine foxo1 mRNA respectively (Supplementary Fig. 1b-c). To determine whether these differences accounted for the differences in phenotype we modified the human lentiviral vector such that expression was driven by a PGK or CMV promoter, both of which have been reported to give lower transgene expression relative to the EF1 α promoter (66). In our hands the CMV promoter drove a significantly lower level of mCherry or FOXO1 expression relative to the EF1 α and PGK promoters (Supplementary Fig. 5f) enabling us to evaluate the impact of high or low FOXO1 gene expression on CAR T cell phenotype. Whilst FOXO1-ADA was able to induce an increase in the CD45RA⁺CD62L⁺ population when driven by the EF1 α , PGK or CMV promoters, the phenotype evoked by wild type FOXO1 was dependent on promoter strength with only a mild phenotype observed using the weak CMV promoter (Supplementary Fig. 5g-h). However, FOXO1-ADA significantly attenuated the production of IFN γ and TNF by CAR T cells regardless of the promoter used to drive its expression (Supplementary Fig. 5i). Given these results we decided to proceed with evaluation of the impact of FOXO1 WT expression driven by the EF1 α promoter.

Other specific points:

5. Lines 77-80 refer to solid tumors but one of the cited references, number 15, is about leukemia.

Response: We thank the reviewer for this observation and have removed this reference accordingly.

6. Please reference the statement in line 358 about “. . . the critical importance of FOXO1 in maintaining CAR T cell stemness . . . “

Response: This statement was referring to the data in the previous section involving CRISPR/Cas9 mediated deletion of Foxo1. To clarify this, we have amended this text as indicated below, which is now located on line 402.

“Given the strong molecular signature of Foxo1 in IL-15 conditioned CAR T cells and the critical importance of Foxo1 in maintaining CAR T cell stemness as demonstrated through CRISPR/Cas9 mediated targeting”.

7. In presenting figure 1a, the authors never mention the use of IL-7 in addition to IL-15 or IL-2.

Response: We did not refer to IL-7 as this was common to both culture systems i.e. IL-15 and IL-7 or IL-2 and IL-7 and therefore the key comparison is the differential effects of IL-2 and IL-15. However, to avoid confusion we have amended the following text on line 369.

“Previous studies by both our group and others have shown that CAR T cells generated with IL-7 and IL-15 display a less differentiated T cell phenotype and elicit greater long-term persistence relative to CAR T cells cultured in IL-2 and IL-7.”

8. Supplementary figure 1a-b. This is pretty low-level overexpression. Please comment and please show the controls for each transcription factor.

Response: In the process of answering point 4 above we have replaced this with new qRT-PCR data which we believe is a more accurate representation of the extent of overexpression.

9. In the model in figure 2g-i don't the CAR-T cells attack the endogenous HER2 expressing tissues?

Response: This syngeneic model enables us to identify if CAR T cells target endogenous Her2 expressing tissues, which in this case is predominantly the cerebellum and the mammary glands of these human Her2 Tg mice (PMID 21098715). We do not observe that the CAR T cells target Her2 expressing host tissues and believe this is due to the higher level of antigen expression on the tumor cells.

Referring to figure 6 if n=7 in panel a, why aren't there 7 data points in b-e? And why are there 8 data points in some part of panel f, but 7 in other parts?

Response: The data represented in the original **Figure 6A** (Now revised **Figure 6D**) and original **Figure 6F** (revised **Supplementary Figure 10F**) was performed on separate cohorts of mice to the original Figure 6b-e (revised **Figure 6E-I**).

Referee #2 (Remarks to the Author):

CAR-T and Adoptive Cellular Therapy (ACT) continue to emerge as a potentially powerful means to treat cancer. This has been spurred on by initial successes and FDA approvals. However, it is clear that there is still much work to be done in terms of developing this modality. Specifically, CAR-T cells have yet to demonstrate significant efficacy against solid tumors. Likewise, it is clear that efficacy is related to persistence of "memory" like cells which can provide ongoing effector cell generation. In this report the author's provide evidence that engineering cells to over-express FOXO1 can lead to more robust CAR-T cells fit to overcome these hurdles.

Overall, this is a very exciting and important work that will provide great insight to the field. The paper itself is exceedingly clear and well written and the experiments and data are thoughtful and robust. I enjoyed reading this paper and learned a lot. Specifically, the work is very comprehensive in relating gene expression with metabolism with function. There are just a couple of issues that it would be nice for the authors to address:

Response: We thank the reviewer for the positive appraisal of our work.

First, there is a lot of functional data comparing the Foxo1-ADA with Tcf7. However, it would be nice to see some of the mechanistic comparisons between the two. For example, in Figures 3f and 4J & K how does Tcf7 compare? Also, is there any data regarding the Tcf7 overexpression in the human cells?

Response: We thank the reviewer for this question. A similar suggestion was made by reviewer 1 and thus this question is addressed in response to question 1 from reviewer 1.

Second, the differences between the overexpression of the Foxo1-ADA and Foxo1 in the human cells is important. The authors suggest that the differences may be due to higher levels of overexpression as a consequence of using a different promoter. This hypothesis should be tested and the authors should consider performing more experiments to hone in to the cause of the differential effects of the Foxo1-ADA versus the Foxo1 overexpression in the human cells.

Response: We thank the reviewer for this question. A similar suggestion was made by reviewer 1 and thus this question is addressed in response to question 4 from reviewer 1.

Referee #3 (Remarks to the Author):

Summary of the key results: the authors propose FOXO1 constitutive activation/overexpression as a potential strategy to address limitations of CAR-T therapy for solid tumours.

Originality and significance: The importance of FOXO1 to CAR-T functionality has been elegantly described by Klebanoff et al in JCI Insight in 2015 in the context of AKT inhibition, with similar conclusions to this paper about the role of FOXO1 in promoting superior phenotype and functionality in a CD19CAR-T model. JCI Insight. 2017 Dec 7;2(23):e95103.doi: 10.1172/jci.insight.95103.

Response: We thank the reviewer for highlighting this and we agree that referencing this study is warranted. The key difference in the current approach relative to this study is that such changes evoked by AKTi are transient and T cells differentiate normally after transfer into patients. Our new data (discussed below in response to specific comment 6.) shows that one of the benefits of FOXO1 expression is that the phenotype can be imparted more permanently until CAR T cells become activated. Our discussion of this work is located in a new section of the discussion located on line 784.

“Furthermore, a number of studies have explored mechanisms to enable CAR T cells to adopt less differentiated phenotypes such as the use of the homeostatic cytokine IL-15 or small molecule inhibitors e.g. AKT inhibitors (73), PI3Ki (NCT03274219) or epigenetic modifiers e.g. JQ1 (22) to maintain CAR T cells in culture prior to transfer. Notably, AKT inhibition was shown to enhance the localization of FOXO1 to the nucleus and result in the upregulation of FOXO1 target genes. However, whilst small molecule inhibitors such as AKTi can be applied easily to CAR T cells in culture, the disadvantage for such approaches is that the effect on phenotype is transient such that once the CAR T cells are infused into the patient the CAR T cells differentiate and exhaust in a normal manner. Therefore, a gene engineering approach to improve CAR T cell resistance to exhaustion may be preferable.”

Data & methodology: valid, good quality, well presented data

Appropriate use of statistics and treatment of uncertainties:yes

References: in my opinion need to add Klebanoff paper and talk about AKT inhibition wrt FOXO1 as an alternative (non genotoxic) way to effect the same cell 'fitness' as an engineered approach, and to show how it differs/why your approach is better.

Response: Thank you, please refer to the above response.

Specific comments:

1. Figure 2h: shows difference in tumour growth at day 10-12, but in supplementary figure 2 it looks like majority tumours are only transiently controlled and animals all dead by ~Day 28-35. Is this the best we can expect- why do no animals fully reject tumours? Initial CAR expansion controls and then the CARs stop dividing? Or the TME switches them off, irrespective of FOXO?

Response: Results in **Figure 2** and **Supplementary Figure 2** refer to data using FOXO1-ADA expressing CAR T cells in murine syngeneic models. We believe that the extent of tumor growth delay in these experiments is biologically significant given the aggressive nature of these tumor models. Moreover, whilst cures were not observed in the syngeneic models, we note that cures were observed with human CAR T cells expressing FOXO1. We believe this is likely to be at least partly attributable to the more profound phenotype we see in the human CAR T cell system relative to the mouse CAR T cell system, which our new data suggests is due to more efficient overexpression achieved in the human CAR T cell system (please refer to point 4 in response to reviewer 1). To clarify this point we have added additional data (new **Supplementary Figure 10A**) highlighting the individual growth curves of mice treated with human CAR T cells expressing FOXO1.

2. Figure 3g/h and line 660: the inference that FOXO1-ADA makes CAR-T cells LN resident, is this just a feature of better in vivo expansion with FOXO1-ADA rather than being a

reprogramming/trafficking event? What other features of lymphoid resident cells could you test for to help support this statement?

Response: In the murine CAR T cell model, FOXO1-ADA expression does not significantly enhance *in vivo* expansion. We hypothesise that this may be due to the lower expression of FOXO1-ADA in the murine system relative to what was achieved in the human CAR T cell system as outlined in response to point 4 from reviewer 1.

To further investigate this effect, we performed additional experiments where the number of CAR T cells were enumerated in both the draining and non-draining inguinal lymph nodes. This revealed that the numbers of Foxo1-ADA expressing CAR T cells were significantly increased in the draining lymph nodes but not the non-draining lymph nodes, suggesting that CAR T cells were significantly enhanced in the tumour draining lymph node. This data is shown in new **Figure 3I** and is referred to in the following text located on line 520.

“Interestingly, a more comprehensive analysis of the relationship between Foxo1-ADA overexpression and the number of CAR T cells within draining lymph nodes revealed that Foxo1-ADA specifically enhanced the number of CAR T cells at the draining lymph nodes but not the non-draining lymph node from the opposite flank (Fig. 3i). This suggests that Foxo1-ADA expression specifically enhanced the migration and/or expansion of CAR T cells at the draining lymph node.”

We also note that in our original submission we presented data indicating that CCR7 expression is increased in human CAR T cells following FOXO1 expression (presented in revised **Supplementary Figure 6A**). To investigate this further, we interrogated this in the context of scRNA-Sequencing and observed that CCR7 expression was enhanced on a cluster of cells that were expanded following FOXO1 overexpression. This data is shown in revised **Figure 4I** and referred to in the following text on line 6671.

Indeed, comparing FOXO1 expressing CAR T cells within cluster 1 to control CAR T cells within cluster 0 revealed a significant enrichment for genes associated with less differentiated cells and reduced expression of genes associated with glycolysis and exhaustion, corroborating our results from bulk RNA-sequencing (Fig. 4k). Moreover, cluster 1 cells expressed high levels of CCR7, suggesting that these cells have the potential to traffic to lymph nodes, consistent with the phenotype observed with murine FOXO1-ADA expressing CAR T cells (Fig. 4i).

3. Line 583: should this read 'does NOT preclude epigenetic sic..'

Response: We thank the reviewer for this observation and have amended accordingly.

4. Line 606-608: cannot make any inference about CAR-T toxicity from a model with no functioning immune system (NSG) so this should be significantly softened/removed.

Response: We agree with the reviewer that the use of an immunodeficient model is sub-optimal for assessments of safety. To strengthen this aspect, we have added new data related to the safety of Foxo1-ADA expressing CAR T cells. Please refer to point 3 from reviewer 1. We believe the use of language to soften the claims around the safety of human CAR T cells is probably not necessary with the inclusion of the new data from the syngeneic model, but we would be happy to include this if requested by the editors and/or reviewers.

5. Line 693: this is fundamental. Is the difference between murine and human FOXO-ADA due to higher transcripts and expression in human system which disables CAR-T into being unable to perform effector functions? Can you show the data as to whether there were higher levels of transcripts? If this is simply a question of FOXO1-ADA 'dose' then hypothesis should be quite easy to test- to titrate expression to find 'sweet spot' where preserved stemness and function co-exist, as they did in the murine model.

Response: We thank the reviewer for this suggestion, which was also raised by the other reviewers. Please refer to our response to point 4 from reviewer 1.

6. Line 693 and Figure 5: When you use FOXO1 overexpression instead of FOXO1-ADA in the human model, you show that many of the perceived benefits of FOXO1 overexpression are reversed on stimulation. The critical point to address is whether the cells then revert back to desirable phenotypes when stimulation state is removed i.e. does the continued overexpression of FOXO1 confer a 'plastic' state where cells shuttle between memory and effector depending on the presence of activation? If not, and only transient benefit from FOXO1 gene engineering, could this same benefit be more simply effected by using an AKT inhibitor in vitro as per Klebanoff?

Response: We thank the reviewer for this suggestion. To address this, we have performed new experiments where CAR T cells were stimulated with tumour cells and then rested to determine whether they revert to a less-differentiated phenotype when stimulation is removed. These data show that FOXO1 expression results in a higher proportion of naïve and central memory CAR T cells and fewer terminally differentiated effector cells relative to control CAR T cells. This new data is shown in **Supplementary Figure 8A** and is referred to in the following text on line 606.

“Indeed, after activation followed by 1 week of rest FOXO1 expressing CAR T cells maintained higher level of CD45RA⁺CD62L⁺ (naïve) and CD45RA⁻CD62L⁺ (central memory) CAR T cells relative to controls (Supplementary Fig. 8a).”

Furthermore, we compared the transcriptional profile of CAR T cells after a period of rest. This data revealed that FOXO1-overexpressing CAR T cells maintain a negative enrichment for genes associated with exhaustion following stimulation and rest. This new data is shown in **Supplementary Figure 6M** and is referred to in the following text on line 651.

“We next evaluated this in the context of stimulation and resting. In this repeat, we again observed that prior to stimulation FOXO1-expressing CAR T cells exhibited reduced expression of exhaustion-related genes (Supplementary Fig. 6m). Following stimulation through the CAR both control and FOXO1-expressing CAR T cells upregulated these genes such that there was no significant difference between the groups. However, after a period of rest, the negative enrichment for this gene set was restored in FOXO1 expressing CAR T cells, highlighting that these cells are protected from the transcription of genes associated with exhaustion.”

7. Figure 6 legend: cannot say that FOXO1 improves 'persistence' if testing in an NSG model and only have data out to day 12 or so? Maybe you mean expansion?

Response: We agree with the reviewer and have amended the figure legend accordingly and at relevant sections of the text.

8. Line 789: how are you defining 'exhaustion' here? I can't see all the facets of T-cell exhaustion shown/discussed here other than preserved functionality.

Response: We thank the reviewer for this question. Reviewer 1 also raised a similar question with regards to exhaustion phenotypes. Please refer to our answer to answer to point 2 reviewer 1.

9. Figure 6d: lower Tim3/PD-1 in FOXO1 condition - is this the same in the tumour and spleen?

Response: This question refers to the reduced expression of PD-1 and TIM-3 observed in tumor infiltrating CAR T cells (Original **Figure 6D**, data now shown in revised **Figure 6K**) To address this question, we have performed further analyses on splenic CAR T cells. Combining data from 2 experimental repeats there is no significant difference in expression of PD-1 and TIM-3 in the spleen. This is likely because the majority of control CAR T cells in the spleen are already PD-1 and TIM-3

negative. This data is included below as a reviewer only figure but can be included in the manuscript if requested.

Reviewer Only Figure 1 Expression of PD-1 and TIM-3 on CD8⁺ CAR T cells isolated from the spleens of OVCAR3 tumor bearing mice.

OVCAR3 tumor bearing mice were treated with 2×10^6 control or FOXO1-expressing anti-Lewis Y CAR T cells. At day 13 post treatment, CAR T cells were isolated from the spleen and phenotyped for expression of PD-1 and TIM-3. Data is represented as the mean \pm SEM of $n = 10-12$ per group.

10. Figure 6i: patient cells - Is the expression of FOXO1 in cis with CAR? Linked by 2A/IRES? Is there any functional data here to mirror what you have shown in healthy cells?

Response: For the patient cells we used the same constructs as per experiments with healthy donors (shown in **new Supplementary Figure 5A**). To clarify, the cells used for our study were from the original apheresis, i.e. prior to CAR transduction and so we transduced the cells with both the CAR and FOXO1 or mCherry control.

To further analyse the functional impact of FOXO1 overexpression on patient-derived CAR T cells we have performed new experiments showing that FOXO1 overexpression significantly a) modulates the metabolic function of patient-derived CAR T cells *in vitro*, similarly as is the case with cells derived from healthy controls b) enhances the number of CAR T cells in the spleen and tumors of treated mice c) significantly increases the proportion of CD45RA⁺CD62L⁺ CAR T cells *in vivo* and reduces the expression of PD-1 and TIM-3. The Seahorse data is shown in **New Figure 6K**, and the *in vivo* data is shown in **New Figures 6L-N**. This data is referred to in the following text located on line 762.

Consistent with with healthy donor derived CAR T cells, overexpression of FOXO1 in patient-derived CAR T cells led to a significantly enhanced maximal respiratory rate and spare respiratory capacity (Fig. 6k). To confirm that FOXO1 overexpression could modify CAR T cell phenotype and function in vivo, we treated OVCAR-3 tumor bearing mice with CAR T cells derived from 2 individual patients. In both cases, FOXO1 expression in patient-derived CAR T cells significantly enhanced the numbers of CAR T cells in both the spleens and tumors of treated mice (Fig. 6l). Importantly, these CAR T cells also exhibited an increased proportion of CD45RA⁺CD62L⁺ “stem like” phenotype in the spleen (Fig. 6m) and a less exhausted phenotype in the tumors characterized by reduced expression of PD-1 and TIM-3 (Fig. 6n), indicating that that FOXO1 overexpression can similarly modulate patient derived CAR T cells towards a less differentiated state.

Reviewer Reports on the First Revision:

Referees' comments:

Referee #1 (Remarks to the Author):

While I greatly appreciate the new studies that have been performed and the authors efforts to address my, and other reviewer's, comments, substantial and critical problems remain.

I continue to feel that the argument that Foxo1 is relevantly different from Tcf7 is extremely weak. While significant differences in biomarkers are observed (e.g., figures 6b and 6c), I am not persuaded that they translate into anything meaningful. The tumor results in 6a are very similar. The scatter in the Tcf7 numbers prevent them from being significant from control, but I suspect that if the n's were higher, they would achieve significance. I am also disturbed that in figures 4g and 4h, which are cited as differentiating Foxo1 and Tcf7, that the two were not directly compared but were obviously done in different experiments. That really makes it impossible to properly interpret.

The Klebanoff paper from JCI Insight presents a separate but possibly more important issue as it dramatically undercuts the significance of this work. The discussion of why the current data are different is technically correct, but I don't see that they have shown an important new biological principle.

Referee #2 (Remarks to the Author):

The authors have answered my questions by providing new data.

Referee #3 (Remarks to the Author):

I am happy that we queries have been addressed by the authors.

Author Rebuttals to First Revision:

Referee #1 (Remarks to the Author):

Referee: While I greatly appreciate the new studies that have been performed and the authors efforts to address my, and other reviewer's, comments, substantial and critical problems remain.

Response: We are glad that the reviewer appreciates the new studies performed in our previous resubmission. We hope that our revised manuscript addresses any remaining reservations of the reviewer.

Referee: I continue to feel that the argument that Foxo1 is relevantly different from Tcf7 is extremely weak. While significant differences in biomarkers are observed (e.g., figures 6b and 6c), I am not persuaded that they translate into anything meaningful. The tumor results in 6a are very similar. The scatter in the Tcf7 numbers prevent them from being significant from control, but I suspect that if the n's were higher, they would achieve significance.

Response: We would like to note that the experiment referred to by the reviewer in **Figure 6a** was not specifically designed to compare the therapeutic effect of FOXO1 and TCF7-expressing CAR T cells.

Rather, in this experiment, tumors were collected at a specific time point (prior to when differences in therapeutic activity are most obvious) because this, in our opinion, provided a more meaningful way to address the reviewer's initial question with regards to the comparison of FOXO1 and TCF7 on CAR T cell phenotype and function *in vivo*.

Given the lack of conclusive data on the impact of TCF7 on the therapeutic activity of CAR T cells in this experiment we have amended the text in the results section on **line 470** as follows:

Previous text: "To compare the impact of TCF7, ID3 and FOXO1 overexpression *in vivo* we treated OVCAR-3 tumor bearing mice with CAR T cells expressing each transcription factor. Strikingly FOXO1 was the only transcription factor that enhanced CAR T cell efficacy....."

New text: "To compare the impact of TCF7, ID3 and FOXO1 overexpression on CAR T cell phenotype and function *in vivo* we treated OVCAR-3 tumor bearing mice with CAR T cells expressing each transcription factor. FOXO1 overexpression enhanced CAR T cell efficacy....."

For the reviewer's interest we have subsequently performed a comparison of the therapeutic activity of FOXO1- and TCF7-expressing CAR T cells using a suboptimal dose of 2e5 CAR T cells. Under these conditions, FOXO1-expressing CAR T cells elicit a significantly enhanced therapeutic effect relative to TCF7-expressing CAR T cells. This is included below as a reviewer only Figure, but can be included in the manuscript upon request.

Reviewer Only Figure 1: FOXO1 overexpression enhances the therapeutic activity of anti-Lewis Y CAR T cells

OVCAR3 tumor bearing mice were preconditioned with 1Gy total body irradiation and then treated with 2×10^5 of indicated CAR T cells. Data represents the mean \pm SEM of 6 mice per group from a representative experiment of $n = 2$. **** $p < 0.0001$, two way ANOVA.

Furthermore, we also note that in Dr. Weber's study (also under consideration at Nature, available as a preprint: <https://www.researchsquare.com/article/rs-2802998/v1>) they have independently evaluated the impact of TCF7 and FOXO1 on the efficacy of anti-CD19 CAR T cells in the NALM6 model and show enhanced therapeutic activity only in the context of FOXO1 overexpression.

Reviewer Only Figure 2- Data from Figure 5 of Doan *et al.*- Preprint

A subcurative dose of $0.1-0.2 \times 10^6$ tNGFR-purified CD19.28 ζ CAR T cells were infused into Nalm6 leukemia-bearing mice 7 days post-engraftment. Stress test Nalm6 model schematic (left) and survival curve (right) are shown ($P < 0.0001$ log-rank Mantel-Cox test). Data are from 2 donors ($n = 4-5$ mice per condition).

We also wish to provide the following reviewer only Figure that we believe further clarifies the differences evoked by FOXO1 and TCF7. It may not have been apparent from the data presented in **Figure 4b** that the impact of FOXO1 on the emergence of a CD45RA⁺CD62L⁺ phenotype is profoundly different to that of TCF7. Below we present data from a matched donor and paired analysis from 7 individual donors highlighting the significant difference in the effect mediated. These data can also be included in the manuscript upon request.

A.

B.

Reviewer Only Figure 3 Expression of CD45RA and CD62L on CD8⁺ CAR T cells

A. FACS profile of a representative donor and B. paired analysis of n = 7 donors at day 14 post transduction. ****p<0.0001, one way ANOVA.

Referee: I am also disturbed that in figures 4g and 4h, which are cited as differentiating Foxo1 and Tcf7, that the two were not directly compared but were obviously done in different experiments. That really makes it impossible to properly interpret.

Response: We acknowledge this concern. As the reviewer is aware the Seahorse experiments for TCF7 expressing CAR T cells were added as additional experiments following the reviewer's initial comments, so we did not perform these head-to-head with FOXO1 expressing CAR T cells. Given this concern, we have removed the Seahorse data pertaining to TCF7 expressing cells (**Figure 4h**) and removed references to a direct comparison on the metabolic phenotype evoked by TCF7 and FOXO1. Therefore, the following sentences have been removed/ amended in the results and discussion sections respectively.

Text removed from the Results:

“Notably, there were minimal changes to these parameters in the context of TCF7 overexpressing CAR T cells, highlighting that FOXO1 overexpression was a superior strategy to enhance the metabolic fitness of CAR T cells (Fig. 4h).”

Previous Discussion text:

“Further comparison between FOXO1- and TCF7- overexpressing CAR T cells revealed that FOXO1 was unique in its ability to favorably modulate metabolic function and was significantly more able to drive transcriptional changes consistent with a more persistent and less exhausted CAR T cell product.”

New Discussion text (line 613):

“Further comparison between FOXO1- and TCF7- overexpressing CAR T cells revealed that FOXO1 was significantly more able to drive transcriptional changes consistent with a more persistent and less exhausted CAR T cell product.”

The Klebanoff paper from JCI Insight presents a separate but possibly more important issue as it dramatically undercuts the significance of this work. The discussion of why the current data are different is technically correct, but I don't see that they have shown an important new biological principle.

Response: As outlined in our previous response to referee #3, the key difference in the current approach relative to this study is that such changes evoked by AKTi are transient and T cells differentiate normally after transfer into patients. Our new data shows that one of the benefits of FOXO1 expression is that the phenotype can be imparted more permanently until CAR T cells become activated by encountering antigen.

Referee #2 (Remarks to the Author):

The authors have answered my questions by providing new data.

Referee #3 (Remarks to the Author):

I am happy that we queries have been addressed by the authors.

Response: We are pleased that Reviewer's 2 and 3 are satisfied by our new data.

Author Rebuttals to First Revision:

Referees' comments:

Referee #1 (Remarks to the Author):

I appreciate the detailed response and the inclusion of new data in the rebuttal letter. I believe the manuscript would be significantly strengthened by including Reviewer Only Figure 1 and Reviewer Only Figure 3 in the paper. They are important to support the authors conclusions.